# Gonadotrophs have a dual origin, with most derived from early postnatal pituitary stem cells

Daniel Sheridan[1], Probir Chakravarty[2], Gil Golan [3], Yiolanda Shiakola[1], Jessica Olsen [4], Elise Burnett[1], Christophe Galichet[1], Tatiana Fiordelisio [5], Patrice Mollard [6], Philippa Melamed [3], Robin Lovell-Badge [1] ✉ & Karine Rizzoti [1] ✉

Gonadotrophs are the essential pituitary endocrine cells for reproduction. They produce both luteinizing (LH) and follicle-stimulating (FSH) hormones that act on the gonads to promote germ cell maturation and steroidogenesis. Their secretion is controlled by the hypothalamic gonadotrophin-releasing hormone (GnRH), and gonadal steroid feedback. Gonadotrophs first appear in the embryonic pituitary, along with other endocrine cell types, and all expand after birth. While gonadotrophs may display heterogeneity in their response to GnRH, they appear, at least transcriptionally, as a homogenous population. The pituitary also contains a population of stem cells (SCs), whose contribution to postnatal growth is unclear, in part because endocrine cells maintain the ability to proliferate. Here we show an unsuspected dual origin of the murine adult gonadotroph population, with most gonadotrophs originating from postnatal pituitary stem cells starting early postnatally and up to puberty, while embryonic gonadotrophs are maintained. We further demonstrate that postnatal gonadotroph differentiation happens independently of gonadal signals and is not affected by impairment of GnRH signalling. The division of gonadotrophs based on separate origins has implications for our understanding of the establishment and regulation of reproductive functions, both in health and in disease.

The reproductive axis comprises the hypothalamus, the pituitary gland and the gonads. While all its components are assembled in the embryo, it only starts to be active postnatally, initially during a transient period known as minipuberty, which is important for future male fertility and cognitive development in both sexes[1–3]. Activity will start again at puberty, which marks the onset of reproductive capacity. The hypothalamic gonadotropin-releasing hormone (GnRH) has a central regulatory role during these periods[4]. Its pulsatile secretory patterns are regulated by a neuronal network that integrates peripheral signals; it is released in the hypophyseal portal system, through which it reaches the pituitary where it controls production and secretion of luteinizing (LH) and follicle-stimulating (FSH) hormones. The two gonadotrophins act in turn on the gonads, regulating steroid and gamete maturation. Gonadal steroids exert crucial feedback both at the hypothalamic and pituitary level.

[1]Laboratory of Stem Cell Biology and Developmental Genetics, The Francis Crick Institute, London NW1 1AT, UK. [2]Bioinformatics core, The Francis Crick Institute, London NW1 1AT, UK. [3]Faculty of Biology, Technion Israel Institute of Technology, Haifa 32000, Israel. [4]Genetic Modification Service, The Francis Crick Institute, London NW1 1AT, UK. [5]Laboratorio de Neuroendocrinologia Comparada, Laboratorio Nacional de Soluciones Biomimeticas para Diagnostico y Terapia, Universidad Nacional Autonoma de Mexico, Mexico City, Mexico. [6]Institut de Génomique Fonctionnelle, University of Montpellier, CNRS, Inserm, 34094 Montpellier, France. ✉e-mail: robin.lovell-badge@crick.ac.uk; karine.rizzoti@crick.ac.uk

Most adult pituitary gonadotrophs produce both LH and FSH. However, the patterns of their secretion differ; LH secretion closely follows GnRH pulses, while FSH does not, and it is regulated by additional peptide hormones. The modalities of LH and FSH regulation are not completely understood, in particular how secretion of LH and FSH is differentially regulated by the same ligand[5]. Adding to this complexity, we reveal here that gonadotrophs comprise a dual population based on different developmental origins.

Gonadotrophs initially arise in the embryonic pituitary. They are amongst the first endocrine cell types to commit, at 12.5dpc in the mouse, in the ventral pituitary primordium, with upregulation of the glycoprotein α, the subunit common to LH, FSH and TSH (thyroid stimulating hormone). LHβ and FSHβ subunits are only expressed from 16.5dpc, which is considered the true gonadotroph birthdate, and GnRH is involved in this, at least in males[6]. Postnatally, gonadotrophs expand dorsally through the gland as a second wave during the first two weeks after birth[7–9].

We and others have shown that the pituitary contains a population of stem cells (SCs)[10]. These do not play a significant role during normal turnover in the adult gland. However, their contribution to the gland's rapid growth postnatally could not be defined because of inefficient genetic lineage tracing tools and use of tamoxifen, a selective estrogen receptor modulator which perturbs normal physiology[11,12]. Here, we developed an efficient and more physiologically neutral lineage tracing system using a doxycycline-dependant $Sox2^{rtTA}$ tool and uncovered that the majority of gonadotrophs differentiate in both sexes from postnatal pituitary SCs, during a period encompassing minipuberty, starting early postnatally and up to puberty. However, their differentiation does not depend on gonadal feedback or physiological GnRH levels. These gonadotrophs invade the gland from the SC niche, with the exception of a small ventral domain where embryonic gonadotrophs remain confined. The discovery of a dual origin for gonadotrophs may help understand aspects of gonadotrophin regulation and mechanisms of diseases affecting puberty and fertility.

## Results

### Endocrine cell type-specific mechanisms underline postnatal expansion

In mouse and humans, all the 6 different pituitary endocrine lineages, somatotrophs secreting growth hormone (GH), lactotrophs secreting prolactin (PRL), thyrotrophs secreting TSH, corticotrophs secreting Adrenocorticotrophic hormone (ACTH), melanotrophs secreting melanotroph-stimulating hormone (MSH) and gonadotrophs, emerge in the embryonic pituitary[13,14]. In the mouse, this is followed postnatally by a period of significant growth until the gland reaches its mature size[15,16], following both an increase in cell numbers, due to proliferation of both stem and endocrine cells[17–19], and endocrine cell swelling, as the secretory apparatus matures[20]. Along with SCs, a population of POU1F1[+ve] hormone[-ve] progenitors, committed to lactotroph, somatotroph or thyrotroph fate, has also been reported[19]. To better characterise this phase, we examined how proportions of each endocrine population evolve, by performing automated cell counting on dispersed pituitary cells stained for the hormone they secrete, at P5, P12, P21, 7 weeks and one-year-old in both males and females (Fig. 1A, Supplementary Table 1). At 7 weeks, the percentage of endocrine cells obtained for each population (Fig. 1A) matches those described previously in adults[21]. From P5 to adulthood, the proportion of somatotrophs and lactotrophs increased the most, leading to their known sex-specific dominance in the male and female pituitaries, respectively (Fig. 1A). Gonadotrophs also exhibit significant postnatal growth during the three first postnatal weeks. In contrast, the proportions of corticotrophs and thyrotrophs, which are near their highest at birth, either remain stable or decrease as the gland matures. To relate this to the gland growth, we counted the total number of pituitary cells

(Supplementary Fig. 1A, Supplementary Table 2); we observe that the female pituitary, known to be larger and heavier than the male one, contains significantly more cells from 7 weeks. We then estimated the number of cells for each endocrine type using our previous counts (Fig. 1A). These data (Fig. 1B) suggest that all endocrine cell numbers increase, except thyrotrophs, which remain similar. To better visualize how each population evolves, we related cell-type proportions to their initial percentage at P5 (Fig. 1C). In both sexes lactotrophs prominently increase in agreement with their mainly postnatal expansion. Representing the second largest increase, both gonadotrophs and somatotrophs display a similar pattern of expansion, undergoing an approximately 3-fold increase in proportion.

To investigate mechanisms underlying populations expansion, we assessed cell proliferation during the first three weeks post-birth by performing one-hour EdU pulses (Fig. 1C, Supplementary Fig. 1B, Supplementary Table 3). From P5 to P21, somatotrophs, lactotrophs and POU1F1[+ve];GH/PRL/TSH[-ve] progenitors had the highest percentage of proliferative cells, which correlates with the expansion of these cell types and previous observations[17–19]. In contrast, gonadotrophs, despite the remarkable increase in their population size, had the lowest proliferation rate of all endocrine cell types at all assessed time points. Finally, SOX2[+ve] SCs showed high proliferation rates early on, with significantly more proliferation in males at P5 ($P < 0.05$, Table S3), which then declined noticeably between P12 and P21, as reported[17,18].

These results expand previous analyses, highlighting the differential growth of each endocrine population, and show that different mechanisms underlie their evolution. While somatotrophs and gonadotrophs appear to progress at a comparable rate, their different proliferative indices suggest that although division of existing somatotrophs may play a role in their expansion, this is not the case for gonadotrophs.

### Neonate SOX9iresGFP[+ve] SCs are predicted to contribute to post-natal pituitary endocrine cell emergence

To characterise the SCs during their highly proliferative period and assess their contribution to the formation of new endocrine cells, we performed single-cell RNAseq on the SOX9iresGFP[+ve] fraction at post-natal day 3 (P3) in males and females. This fraction comprises SCs and their immediate progeny because GFP persists longer than SOX9, allowing for a short-term tracing of the SCs[22]. Male and female datasets were integrated (Fig. 2A, Supplementary Fig. 2). Uniform Manifold Approximation and Projection (UMAP) allowed clustering of $Ednrb^{+ve}$ and $Aldh1a2^{+ve}$ SCs, representing respectively cleft and anterior lobe SCs[22]. Clusters corresponding to all endocrine cell types were present in the dataset, with gonadotrophs representing the largest endocrine cluster. We also observed a $Pitx1^{-ve}$ $Decorin^{+ve}$ cluster labelled "extra-pituitary origin" (Supplementary Fig. 2); this may contain mural cells of neural crest origin, as we showed in adults[22]. We observe a relatively high proportion of proliferative cells, as expected (Fig. 1C)[17], and the presence of two clusters in which expression of SC markers decreases, but are negative for any endocrine marker, suggesting that they represent differentiating SCs (thus named SC differentiating 1 and 2). These clusters are characterised by expression of the WNT pathway transcriptional activator $Lef1$, suggesting activity of the pathway, which is consistent with the proposed paracrine role of SC-derived WNT on their differentiating progeny[23]. However, the expression of cell cycle inhibitors $Cdkn1a$ and $Gadd45g$ (Supplementary Fig. 2)) suggests that cell proliferation is limited in these progenitors.

We next looked for sex differences in cluster contribution (Fig. 2B). A pairwise comparison for proportion test showed that the gonadotroph cluster comprised more male cells and conversely, that the $Ednrb^{+ve}$ SC cluster had more female cells. This suggests that male SCs differentiate more into gonadotrophs than female SCs at this stage.

Pseudotime analyses were performed using Slingshot (24 (Fig. 2C, D)), and gene regulatory networks (GRN) analysis using SCENIC[24]

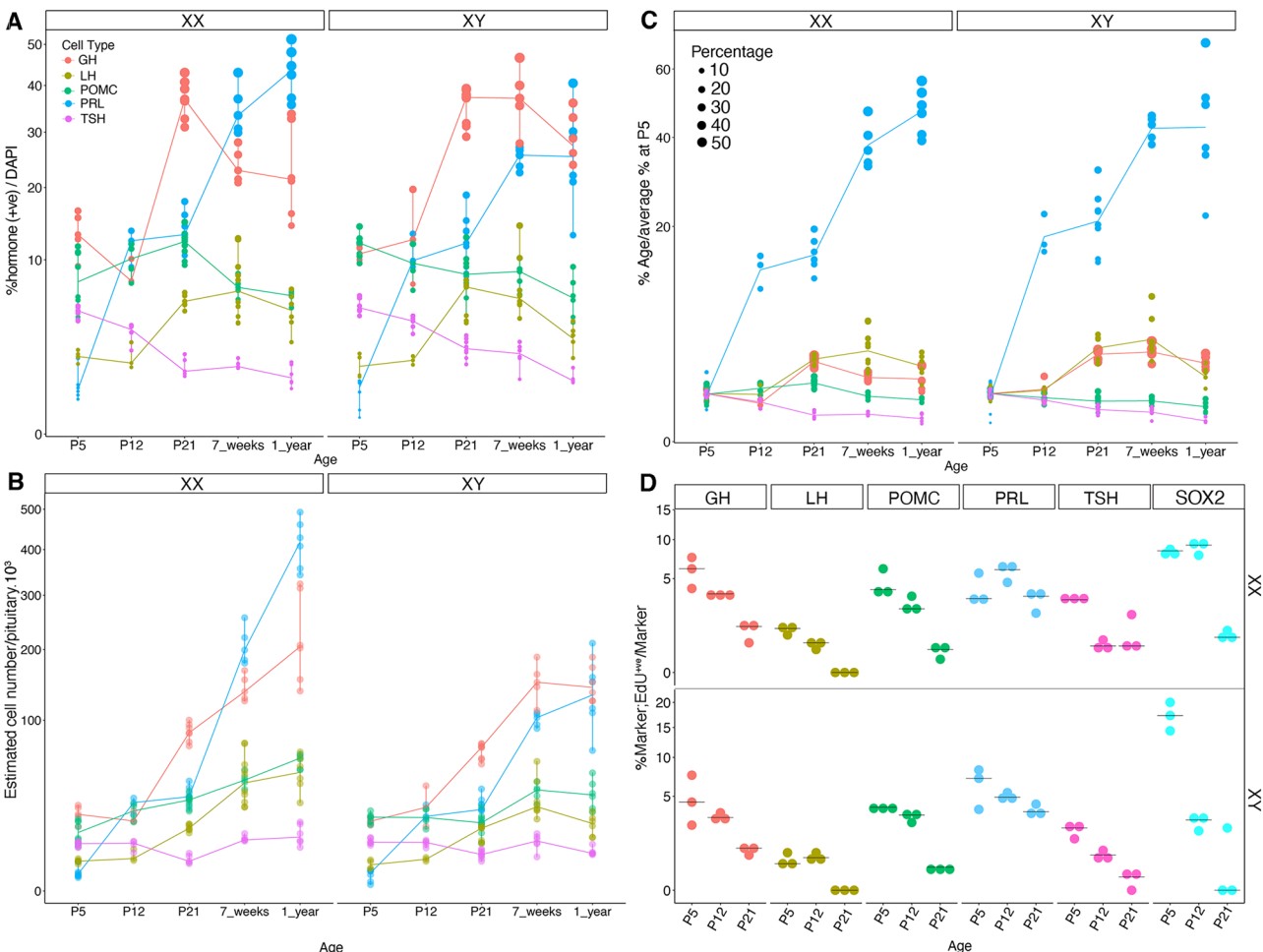

**Fig. 1 | Cell type-specific expansion in the postnatal pituitary. A** Percentages of endocrine cells from dissociated pituitaries stained for each endocrine hormone (GH for somatotrophs, PRL for lactotrophs, TSH for thyrotrophs, LH for gonadotrophs and POMC for both corticotrophs and melanotrophs) from P5 to one-year-old males and females ($n > =3$ pituitaries/age/sex, Supplementary Table S1). **B** Estimated numbers of endocrine cells per pituitary. Total cell numbers were independently counted (Supplementary Fig. 1, Supplementary Table 2) and percentages

assessed in A used to estimate the number of cells for each endocrine population. **C** Evolution of each population: the percentage at each age is shown in relation to the average percentage at P5. The size of dots relates to the proportion of each population in the gland (Supplementary Table S1). **D** Percentage of EdU-positive cells/marker following a one-hour EdU pulse. Each dot represents one pituitary ($n = 3$ pituitaries/age/sex, Supplementary Table S3). Source data are provided as a Source Data file.

(Fig. 2E). The SCENIC analysis identifies regulons consisting of a transcription factor and its putative targets and quantifies their activity according to the expression patterns. Trajectories toward all endocrine cell types were projected (Fig. 2C, Supplementary Fig. 3) with a common root starting from *Ednrb*[+ve] SC then progressing to *Lef1*[+ve] clusters. Given that it was the predominant derivative endocrine cell type, we focussed our attention on the gonadotroph trajectory. These analyses showed known regulators and markers of gonadotroph fate acquisition (*Nr5a1*, *Fshb*, *Gnrhr*, Fig. 2D, E), but also transcripts from genes encoding the bHLH transcription factors *Neurod1*, *4* and *Nhlh2* suggesting that these are likely involved in gonadotroph differentiation as well. In the embryo, *Ascl1;Neurod1/4* deleted embryonic pituitaries show reduced gonadotroph numbers[25] while puberty is impaired in *Nhlh2*[-/-] animals[26]. In the latter, the number of GNRH neurons is reduced but the pituitary is also affected. Our analyses suggest that these factors are involved during the terminal differentiation of gonadotrophs. In addition, our data show that *Foxp2*, a marker of gonadotrophs in the adult[27], is expressed in SCs and progenitors. Its presence in all lineages in our SCENIC analyses, as previously seen in adult pituitaries[28], implies an involvement in endocrine cell fate acquisition (Fig. 2E, Supplementary Fig. 3). Similarly, the presence of *Gli2* in all our GRN analyses

suggest that the Hedgehog (Hh) pathway is active in differentiating SCs. The pathway is important for pituitary embryonic development and is potentially involved in adult pituitary tumor formation[29]. We find that the only Hh pathway ligand to be expressed in our dataset is *Shh*, exclusively present in SC differentiating 2 (Supplementary Fig. 2), while the Shh receptor, Ptch1, is also expressed in the SCs. This suggests that differentiating cells interact with more naïve SCs, where the *Gli2* regulon is predicted to be active. In agreement with a role for the WNT signalling pathway, a *Lef1* GRN is predicted to be involved in all trajectories. The LEF1 protein expression pattern is very dynamic during the first three postnatal weeks. At P4 we mostly see SOX2;LEF1 double[+ve] cells, and our scRNAseq analyses suggest that proliferation is reduced in these cells compared to more naïve SOX2[+ve];LEF1[-ve] SCs (Fig. 2F). We also see LEF1 co-localisation with POU1F1 or FOXL2 (Fig. 2G, H), correlating with its expression in a transient differentiating population, likely giving rise to lactotrophs and gonadotrophs respectively, according to the UMAP (Fig. 2A and Supplementary Fig. 2). In contrast, from P21, LEF1 marks a distinct, SOX2-negative cell population, often in close proximity to SOX2[+ve] cells (Fig. 2F), as previously shown for its transcript, and these have been shown to be highly proliferative[23], at a time where in turn, proliferation of SOX2[+ve] cells diminishes (Fig. 1C).

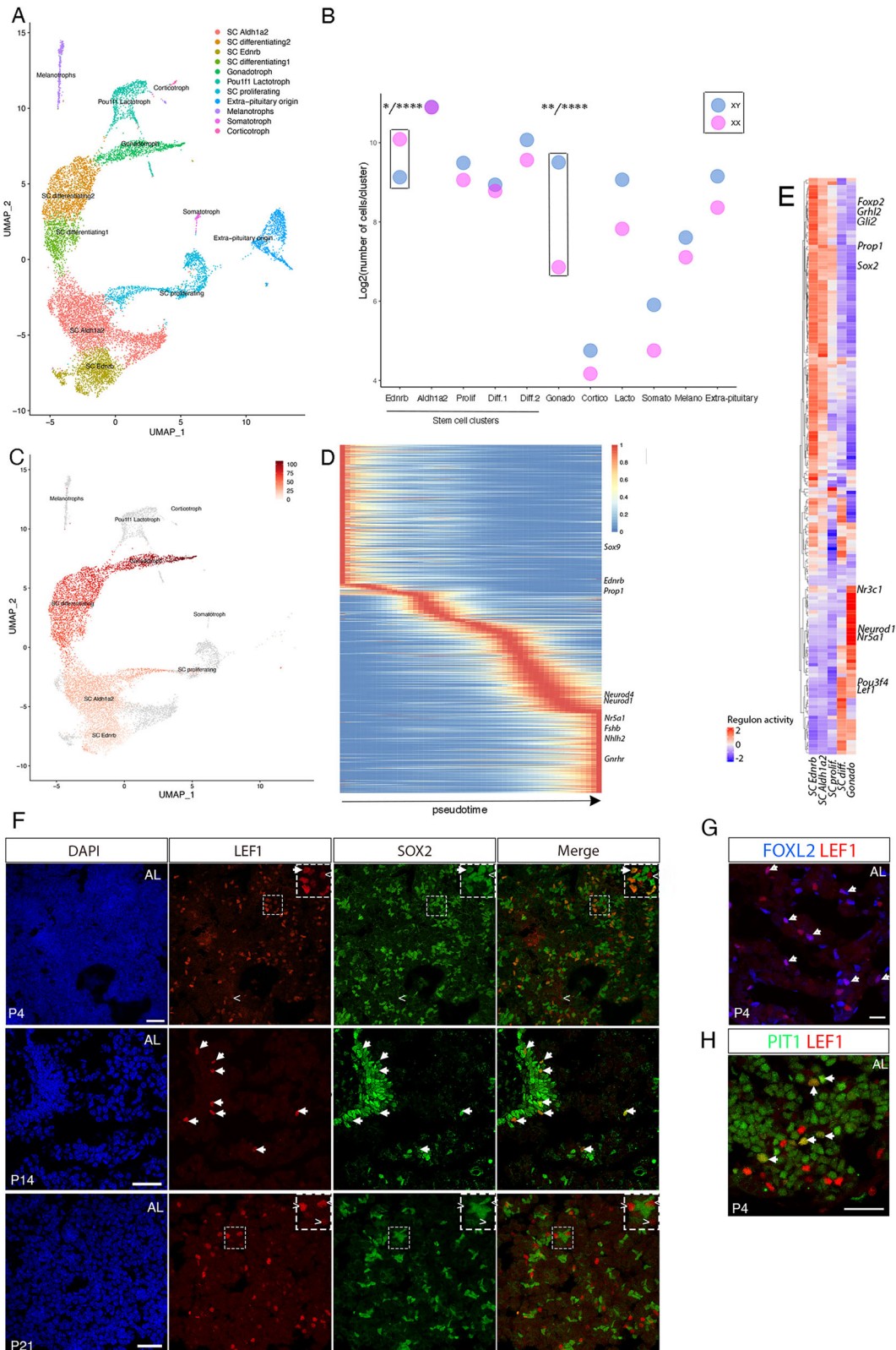

All together, these data suggest that SCs mostly give rise to gonadotrophs at P3. They confirm and predict regulators during this period of high SC activity, with a potential involvement for WNT and SHH signalling.

## SOX2$^{+ve}$ postnatal SCs give rise to most adult gonadotrophs

To examine and expand our transcriptomic analysis predictions, we performed lineage tracing analyses. We generated a $Sox2^{2A-rtTA}$ allele where both copies of $Sox2$ are maintained in order to avoid the hypopituitarism seen in $Sox2^{+/-}$ animals[30] and the use of tamoxifen. We generated $Sox2^{2A-rtTA/+};TetO-Cre;Rosa26^{ReYFP/+}$ ($Sox2rtTA;eYFP$) animals and induced recombination with two 9-Tert-Butyl doxycycline (9TBDox) injections[31], at P0 and P1. 24 hours after the last treatment, approximately 70% of SOX2$^{+ve}$ cells were traced, with all labelled cells expressing SOX2, demonstrating efficiency and specificity of the system (Supplementary Fig. 4). Moreover, treated pups gave rise

**Fig. 2 | Single cell analysis of early postnatal stem cells and their immediate progeny. A** UMAP clustering of integrated P3 male and female SOX9iresGFP[+ve] datasets. **B** Two-sided pair-wise comparison for proportion test with Benjamini Hocberg correction for multiple testing performed to compare male and female cells. Significance is exclusively shown for clusters where distribution was different from all other clusters. This shows that the proportion of Ednrb[+ve] (cleft) SCs is superior in females (p.adj = 1.53.10[-2] to 5.28.10[-60]) while the proportion of differentiating gonadotrophs is higher in males (p.adj = 1.29.10[-3] to 5.83.10[-50], source data are provided as a Source Data file). **C** UMAP representation of the pseudotime gonadotroph trajectory from Ednrb[+ve] SCs to gonadotrophs. Extra-pituitary cells (Pixt1[-ve,] Supplementary Fig. 2) were filtered out. **D** Heatmap showing genes whose expression pattern correlate with the gonadotroph pseudotime trajectory. **E** Heatmap displaying regulons according to the gonadotroph trajectory pseudotime. **F** Double immunostaining for LEF1 and SOX2 from P4 to P21 in male pituitaries. At P4, LEF1 is mostly co-expressed with SOX2 (arrow), with a small number of SOX2[-ve];LEF1[+ve] cells nearby (indicated by <). At P14, LEF1 remains co-expressed with SOX2. By P21, LEF1 no longer colocalises with SOX2. **G, H** At P4, LEF1 is co-expressed in some FOXL2[+ve] (**G**) and POU1F1[+ve] (**H**) progenitors. This experiment was performed at least three times at each timepoint with independent samples. Scale bars represent 30 μm in all panels.

to fertile adults with normal GH and LH levels (Supplementary Fig. 4).

We then analysed the progeny of SCs in *Sox2rtTA;eYFP* pituitaries induced at birth, from P5 and up to one year (Fig. 3A). We observe eYFP;hormone double[+ve] cells for all lineages, as shown previously[11,12]. We quantified SC post-natal contribution to each anterior lobe endocrine population from P5 to one-year old, in both sexes (Fig. 3B, C, Supplementary Table 4). At all ages examined, postnatal SCs contribute very little to somatotroph and thyrotroph populations, both below 1%. A higher proportion of SC progeny is noted in lactotrophs and corticotrophs, ranging from 7 to 12%. Remarkably, comprising up to 80% of eYFP[+ve] cells, and likely more since our lineage tracing is not 100% efficient (Supplementary Fig. 4), we find that gonadotrophs are the population to which SCs contribute by far the most. These results were similar in both sexes.

We then analysed the temporal dynamics of SC-derived endocrine cell emergence (Fig. 3C, Supplementary Fig. 5). For gonadotrophs, from the 10% of the small number of LH;FSH double[+ve] cells present at P5, eYFP[+ve] cell contributions rise sharply to over 40% by P12, surpassing 50% by P21 (pre-weaning), reaching and stabilising at approximately 75% by 7 weeks and up to one year (Fig. 3D and Supplementary Table 5). To further detail the tempo of gonadotroph fate acquisition, we looked at expression of the transcription factor FOXL2, a marker of gonadotroph and thyrotroph commitment[32] (Fig. 3E). At P5, most SOX2[-ve];eYFP[+ve] cells are already positive for FOXL2, while negative for both gonadotroph and thyrotroph hormones, suggesting that cell fate commitment occurs soon after birth. Furthermore, we observe concomitant up-regulation of both LH and FSH from early stages of differentiation in newly differentiated gonadotrophs (Supplementary Fig. 6), while the UMAP suggests a delayed or reduced expression of *Fshb* compared to *Lhb* in differentiating gonadotrophs (Supplementary Fig. 2). To validate our characterisation of LEF1 as a marker of a transient differentiating population, we performed lineage-tracing using *Lef1[CreERT2/+]; Rosa26[ReYFP]* mice, inducing at P0 (Fig. 3F, G). In agreement with our hypothesis, LEF1 progeny contributed to all anterior lobe endocrine cell types (Fig. 3E). Quantification at P12 revealed a proportion of eYFP[+ve] gonadotrophs similar to that found using *Sox2rtTA;eYFP* (Fig. 3F). There was a noticeable increase in eYFP[+ve] somatotrophs and lactotrophs compared to the SOX2 tracing, potentially due to LEF1 being expressed in the POU1F1 lineage, or to the effect of tamoxifen. Additionally, *Lef1* is expressed in thyrotrophs (using the dataset published in[33]), explaining their presence in our quantification.

Finally, to explore the effects of physiological changes during pregnancy, we analysed the SC contribution to all endocrine cell types in lactating dams and age-matched virgin females. No significant change was observed (Supplementary Fig. 7).

In conclusion, most adult gonadotrophs originate from postnatal SCs from the first week of life up to puberty[2]. This result fits with the low proliferation rate of gonadotrophs compared to the other endocrine cell types (Fig. 1C) as their expansion is explained by the differentiation of SC, as we have shown. It is notable that the SCs contribute little to other endocrine populations, and pregnancy does not alter patterns of differentiation. WNT signalling is active and potentially important during cell fate acquisition since all committed progenitors descend from LEF1[+ve] progenitors. Finally, gonadotroph cell fate acquisition occurs rapidly, with a significant proportion of *Sox2rt-TA;eYFP* labelled progeny expressing FOXL2 already at P5.

## Embryonic and postnatal-born gonadotrophs are located in distinct domains

Examination of traced *Sox2rtTA;eYFP* pituitaries at P7 revealed a small, eYFP[-ve], LH[+ve] population confined to the medio-ventral surface of the pituitary (Fig. 4A, upper panel). In contrast, at P21, eYFP[+ve] gonadotrophs are enriched dorsally and seem to emerge from the SCs lining the cleft, while the eYFP[-ve];LH[+ve] population remains confined ventrally (Fig. 4A, lower panel). Several eYFP[-ve] gonadotrophs are present away from the ventral medial region; because the tracing method is not 100% efficient (Supplementary Fig. 4), these are likely to represent non-recombined SC progeny. To quantify the preferential localisation of eYFP[+ve] gonadotrophs, we sectioned the entire pituitary transversely, designating the initial 50% of sections as ventral and the latter half as dorsal (Fig. 4B). By P7, corresponding to an early stage of SC differentiation, the prevalence of LH;eYFP double[+ve] gonadotrophs was comparably low ( <10% of LH[+ve] cells) across both regions. In contrast, by P21, up to 80% of dorsal gonadotrophs were eYFP[+ve], while only approximately 30% of LH[+ve] cells in the ventral region were derived from SCs. This disparity underscores the substantial postnatal differentiation of SCs to gonadotrophs within the dorsal region, predominantly emerging from, or adjacent to the pituitary cleft. This finding correlates well with the pseudotime trajectory, which predicts an *Ednrb[+ve]*, therefore cleft SC origin of gonadotrophs (Fig. 2C)[22]. To investigate this, we generated an *Ednrb[2ArtTA]* allele to exclusively trace cleft SCs. In *Ednrb[2A-rtTA/+];TetO-Cre;Rosa26[ReYF/+]* animals induced at birth, we observe a robust contribution of the progeny to the gonadotroph lineage, demonstrating at least a partial cleft-lining origin of SC-derived gonadotrophs (Supplementary Fig. 8). These may invade the pituitary later; conversely, more ventrally located parenchymal SCs may undergo differentiation, explaining the presence of SC-derived gonadotrophs in the ventral half of juvenile glands.

The presence of early eYFP[-ve];LH[+ve] cells in induced *Sox2rtTA;eYFP* pituitaries suggests that these cells have differentiated in the embryo[6]. To assess this, we performed lineage tracing at 11.5dpc in *Foxl2[CreERT2/+];Rosa26[ReYFP/+]* embryos and performed whole mount immunofluorescence in adults, in parallel with *Sox2rtTa;eYFP* pituitaries induced at birth (Fig. 4C). Despite a relatively poor induction efficiency with *Foxl2[CreERT2]*, we observe that eYFP;FSH double[+ve] cells remain exclusively located near the medial-ventral surface of the anterior pituitary in contrast with the post-natal *Sox2rtTa;eYFP* gonadotrophs present throughout the gland.

In conclusion, neonate gonadotrophs emerge at least in part from cleft SCs and populate the entire gland with the exception of a small ventromedial domain exclusively populated by embryonic-born gonadotrophs, which persist into adulthood in both sexes.

## Differentiation of postnatal gonadotrophs is dictated by the physiological context but is independent of gonadal signals and unaffected by impairment of GnRH signalling

To characterise mechanisms underlying predominant SC differentiation into gonadotrophs, and its temporal restriction to the

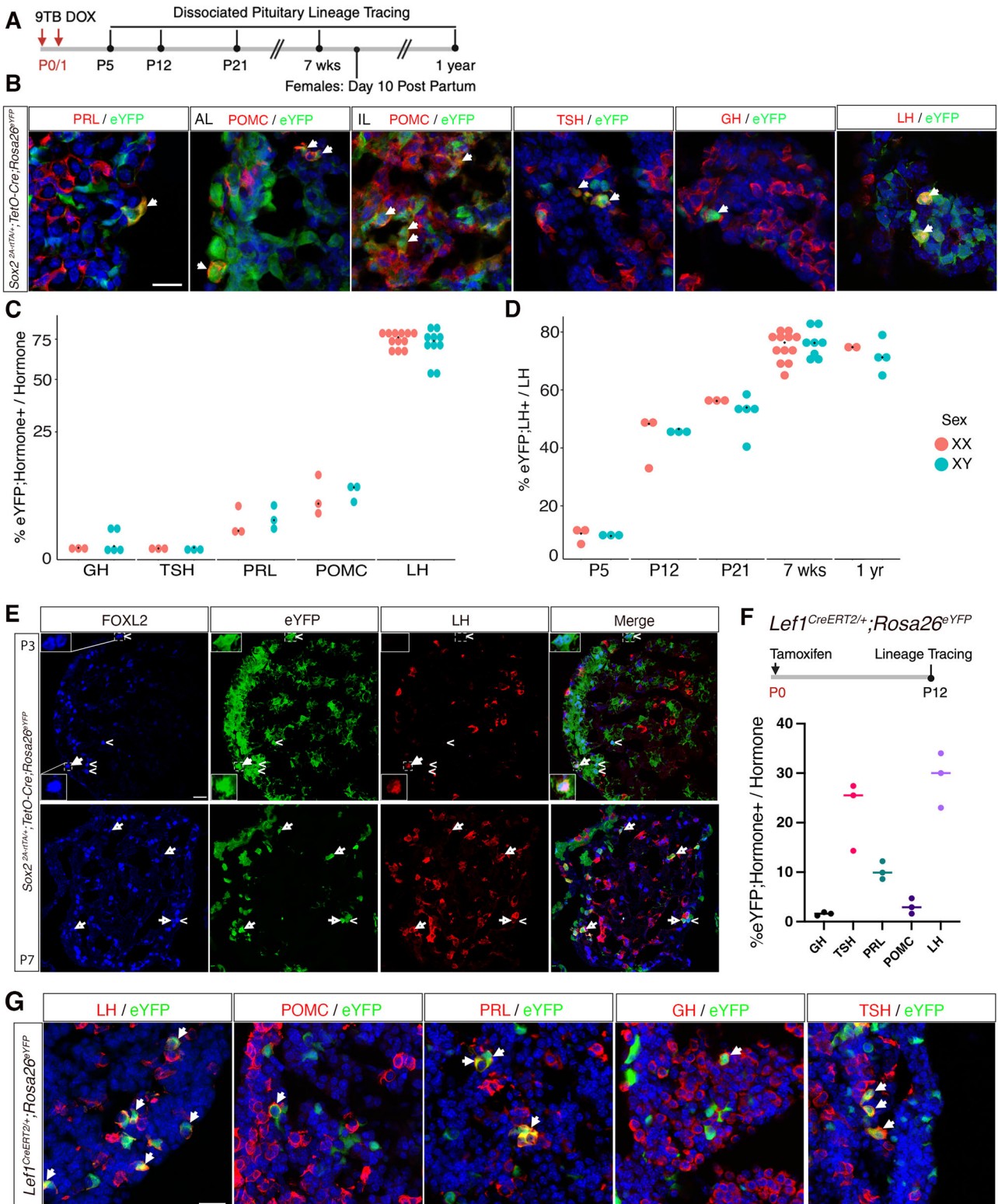

early postnatal period, we first performed pituisphere assays to assess whether neonate SCs were primed toward gonadotroph fate acquisition. We differentiated in parallel neonate and adult pituispheres, as we previously described[34]. We did not observe any preferential differentiation into gonadotrophs for postnatal spheres. Instead, cells of both stages gave rise to LH and POU1F1 lineage cells in similar proportions (Fig. 5A, B). This suggests that neonatal gonadotroph differentiation is dictated by context.

In primates, the abrupt cessation of maternal estrogen negative feedback at birth is proposed to trigger pulsatile GnRH release, inducing minipubertal activation of the HPG axis[1]. We thus examined a role for GnRH in inducing gonadotroph differentiation. SCs do not express its receptor, but we reasoned that embryonic gonadotrophs may relay a signal. We inhibited GnRH action by treating Sox2rtTa;eYFP pups from P0 to P33 with Cetrorelix (Fig. 5C). Long-term treatment with this GnRH antagonist has been shown to further induce a down-regulation of Gnrhr in rats[35]. Blockade of the HPG axis was monitored by

**Fig. 3 | SOX2⁺ᵛᵉ SC and LEF1⁺ᵛᵉ progenitors give rise to most adult gonadotrophs from early postnatal stages up to puberty. A** Timeline of postnatal lineage tracing induction in *Sox2²ᴬ⁻ʳᵗᵀᴬ/⁺;TetO-Cre;Rosa26ᴿᵉʸᶠᴾ/⁺* and harvest timepoints for quantification. **B** Immunofluorescence for lineage-traced eYFP⁺ᵛᵉ SCs and their progeny with each pituitary hormone. SC progeny contributes to all endocrine lineages, as exemplified by eYFP co-localisation with each hormone in a 7-week-old male. **C** Quantification of SC contribution to each cell type in both sexes in 7-week-old mice, as measured by the percentage of eYFP;hormone double⁺ᵛᵉ cells in the total hormone⁺ᵛᵉ population counted in dissociated pituitaries (*n* > =3pituitaries /age/sex Supplementary Table 4). **D** Time course of postnatal SC contribution to the gonadotroph population from P5 to 1 year of age in both sexes (*n* > =3 pituitaries/age/sex). There is no difference between sexes in SC contribution to any cell type (*P* > 0.05, multiple two-tailed unpaired t-test with Benjamini-Hochberg post hoc test, Supplementary Table 5). **E** Immunofluorescence for lineage-traced eYFP⁺ᵛᵉ SCs and gonadotroph markers FOXL2 and LH. SCs rapidly commit to the gonadotroph lineage postnatally, with numerous eYFP;FOXL2 double⁺ᵛᵉ;LH⁻ᵛᵉ cells present at P3 (top), ( < ). From P7 onwards (bottom), most eYFP;FOXL2 double⁺ᵛᵉ cells are also LH⁺ᵛᵉ, (arrow). **F** Timeline of lineage tracing induction in *Lef1ᶜʳᵉᴱᴿᵀ²/⁺; Rosa26ᴿᵉʸᶠᴾ* mice, with tissue counts performed at P12 (*n* = 3 pituitaries). **G** LEF1⁺ᵛᵉ cell progeny contributes to all endocrine cell types, as evidenced by the co-localisation of eYFP with all hormones in a P12 male. Scale bar = 20 μm for all panels. Graphs show individual data points (each dot represents one animal) and group median. AL anterior lobe, IL intermediate lobe. Source data are provided as a Source Data file.

examining gonads when pituitaries were harvested. We observed a clear gonadal size reduction in both sexes and a reduction in CYP17A1 staining in agreement with a downregulation of steroidogenesis in testes (Supplementary Fig. 9), consistent with reduced gonadotrophin levels following GnRH signalling blockade. While we cannot exclude that some GnRH remains active, the effects we observe are consistent with an efficient impairment of its action. In contrast, differentiation of SCs into gonadotrophs was not prevented and we observed a similar proportion of SC-derived gonadotrophs in treated and control animals (Fig. 5C–F). We next investigated a role for gonadal steroids, because suppression of steroid feedback following pituitary target organ ablation affects pituitary adult SC activity[11]. We performed gonadectomies in P3 *Sox2rtTa;eYFP* pups shortly before inducing lineage tracing at P4 and harvested pituitaries in 21-day-old animals (Fig. 5G). In control animals, the proportion of traced gonadotrophs was lower when we induced at this later timepoint (Fig. 5J) compared to previous experiments when we induced just after birth (Fig. 3C, D). This may be due to silencing of the Cre transgene, as this has been reported to happen to tetracycline-controlled cassettes[31]. However, the proportion of eYFP;LH double⁺ᵛᵉ gonadotrophs was not affected by gonad ablation (Fig. 5H–J). In order to block effects of the neonatal male testosterone surge that occurs hours after birth, we used flutamide, an androgen-receptor antagonist[36]. 18.5dpc pregnant dams and newborns were dosed with flutamide, and treatment pursued every 3 days. Gonadotroph emergence at P21 was not affected (Fig. 5K–M). Altogether, these results suggest that neonatal SC differentiation into gonadotrophs is not influenced by gonadal signals and unlikely to depend on GnRH activity. Despite impairment of the HPG axis in both experimental settings, SCs give rise to gonadotrophs in normal numbers and timing.

## Discussion

We demonstrate here that gonadotrophs have a dual origin, with the predominant proportion generated from early postnatal resident SCs up to puberty, and a smaller population of cells born in the embryo persisting in the adult gland. This heterogeneity is furthermore apparent in the gland with both populations occupying different domains: postnatal gonadotrophs emerge dorsally, in agreement with previous observations[7–9] and eventually invade the whole gland except for a small ventral region where cells specified in the embryo remain confined. This is evidence of the existence of two distinct subpopulations of gonadotrophs in the mouse. This phenomenon may extend to primates, because a sharp postnatal increase in gonadotroph numbers is observed in rhesus monkeys[37], suggesting its relevance to human physiology. While single-cell transcriptomic analyses have so far failed to reveal subpopulations of gonadotrophs[38], heterogeneity in response to GnRH has been reported, although with an unknown basis[39]. While mechanisms underlying differential regulation of gonadotrophins secretion have been described[5], the intricacies of LH and FSH differential regulation in response to the same ligand, GnRH, are not yet completely understood, and manipulation of oestrogen hypothalamic feedback on the pulse generator hints at a further level

of gonadotrophin regulation at the pituitary level[40]. Thus, a dual origin and/or distinct localisation for gonadotrophs could explain aspects of their function and regulation. The existence of this dual population may also have implications for our understanding of a range of diseases affecting puberty, either delaying or preventing it, such as in patients affected by congenital hypogonadotropic hypogonadism (CHH), where GnRH signalling is impaired, or when it is induced precociously[1]. For CHH patients, minipuberty has been suggested to represent a window of opportunity for early diagnosis and preventive treatment of pubertal disorders. Our findings give further support to the possibility of early postnatal intervention since this is when most gonadotrophs emerge.

Pituitary SCs can generate all pituitary endocrine cell types[11,12]. However, we show here that they predominantly generate gonadotrophs during the pituitary growth phase, from the first week of life up to puberty[2,3]. Since neonate cells do not show this bias in vitro, this suggests that the physiological context induces this event. We set to test whether GnRH is involved since it initiates minipuberty, reasoning that embryonic gonadotrophs, the only pituitary cells expressing its receptor, may relay a signal to SCs. While GnRH signalling may not be completely abolished in our experiments, the consequences of the antagonist treatment on gonadal development and morphology demonstrate an efficient inhibition of its action. This implies that GnRH is not a crucial factor for differentiation of pituitary SCs in gonadotrophs, in agreement with previous observations implying the presence of gonadotrophs in the absence of GnRH signalling. In CHH patients, fertility can indeed be restored by GnRH administration[1] and some *Gnrhr* mouse mutants can be rescued by pharmacological chaperones[41] demonstrating the presence of gonadotrophs. While GnRH signalling is involved during embryonic gonadotroph emergence[6], our results suggest that the gonadotrophs observed in animal models that lack GnRH signalling, contain this postnatal SC-derived population. We then tested whether gonadal feedback played a role by performing neonatal gonadectomies. In adults, removal of a pituitary target organ induces increased secretion, and sometimes numbers and differentiation from SCs of the cell type normally regulating the ablated organ[11,42]. However early postnatal gonadal ablation did not alter differentiation of neonate SCs into gonadotrophs in either sex, nor did perinatal onset of testosterone antagonism in males. This suggests that other factors specific to the neonate physiological context are implicated, maybe involving interactions with other pituitary endocrine populations, as recently shown between corticotrophs and somatotrophs[43].

Gonadotrophs are the only pituitary endocrine population displaying such a clear dual origin. Since we did not observe a significant contribution of postnatal SCs to any other population, this implies that their expansion is explained by amplification and differentiation of committed progenitors, and proliferation of differentiated cells. In agreement, in the POU1F1 lineage, we observe proliferation of both POU1F1⁺ᵛᵉ;hormone⁻ᵛᵉ progenitors and POU1F1⁺ᵛᵉ;hormone⁺ᵛᵉ cells. In the adult, endocrine cells maintain their capacity to divide, and this is enough, at least for

corticotrophs, to ensure their turn-over in physiological conditions[44], without a requirement for stem cells[45].

It is worth highlighting that in addition to originating from embryonic versus postnatal SCs, which both come from Rathke's pouch (RP)[11], other characteristics distinguish these SCs. RP cells express SOX2, but not SOX9, while postnatal SCs express both markers. It remains to be determined whether the properties of their

gonadotroph progeny are influenced by the activity of these factors. While gonadotrophs may be unique in the pituitary with regard to their dual origin, other endocrine cells can show this distinction. Testicular Leydig cells first differentiate in the embryo, however, this pool regresses postnatally to be mostly, but not completely, replaced by postnatal cells[46]. Of interest, postnatal Leydig cells arise at the same time as the postnatal gonadotrophs. Indeed, postnatal gonadotrophs

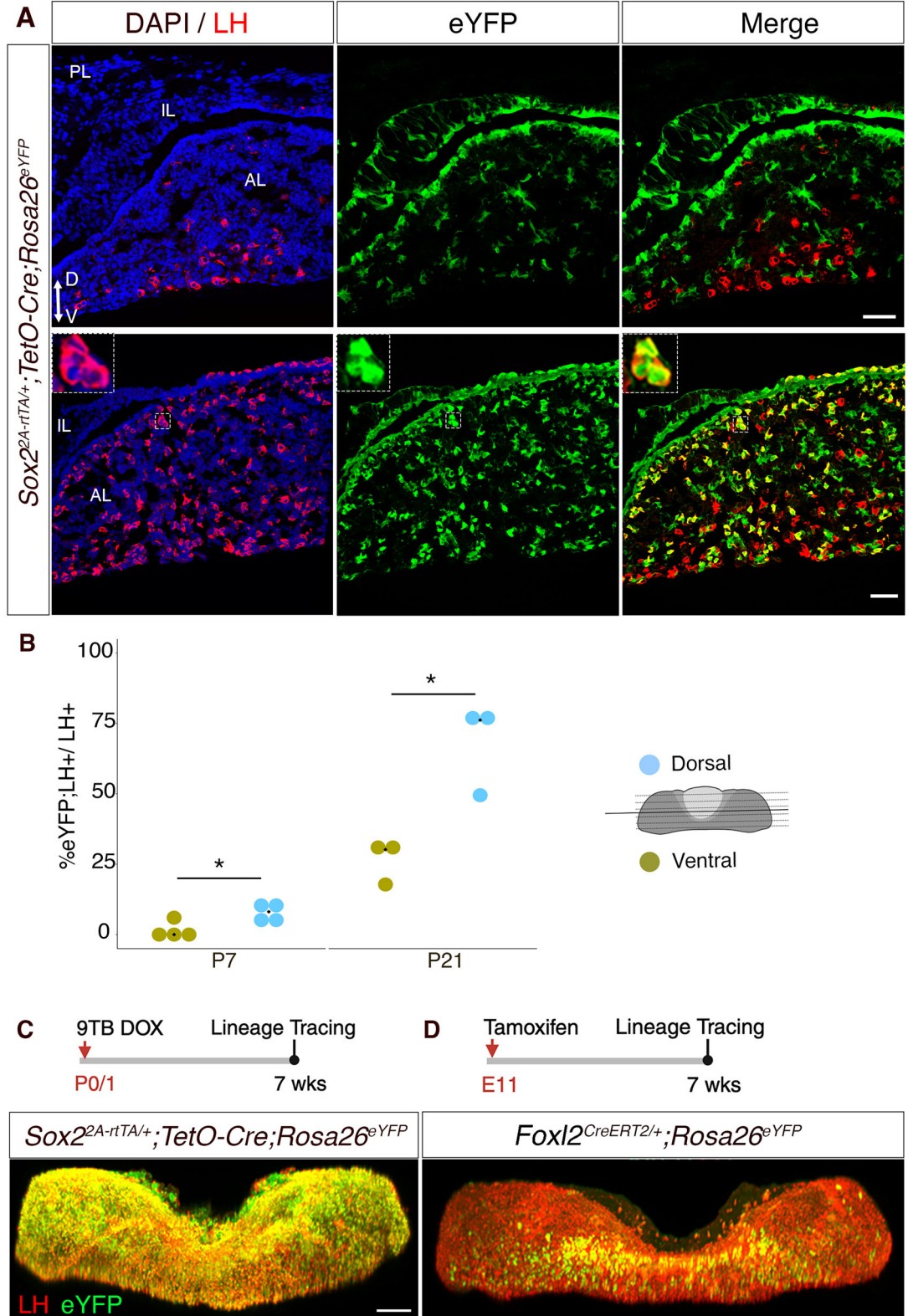

**Fig. 4 | Postnatal and embryonic gonadotrophs occupy different domains in the pituitary. A** Immunofluorescent staining for LH and eYFP showing non-lineage traced, eYFP[-ve];LH[+ve], ventral gonadotrophs at P7 (upper panel) in *Sox2rtTA;eYFP* pituitaries induced at birth. At P21 (lower panel), SC-derived lineage traced eYFP[+ve];LH[+ve] gonadotrophs are enriched dorsally. **B** Quantification of stem cell-derived gonadotrophs (eYFP;LH[+ve]) in ventral (gold dots) and dorsal (blue dots) pituitary regions at P7 (*n* = 4 pituitaries) and P21 (*n* = 3 pituitaries). Pituitaries were sectioned transversely, with the first 50% designated ventral and the rest dorsal. Immunofluorescence cell counts were performed to calculate the percentage of eYFP;LH[+ve] /LH[+ve] per region. Each dot represents one animal, and the diamond indicates the median value. Data from a minimum of *N* = 3 female pituitaries with at least 700 gonadotrophs counted per animal/pituitary. Similar patterns were observed in males (unpaired two-tailed t-test after angular transformation, *p* = 0.0204 for P7 and p = 0.0166 for P21). Source data are provided as a Source Data file. **C, D** Whole-mount eYFP and FSH double immunofluorescence on a *Sox2rt-TA;eYFP* pituitary induced at birth (**C**) and a *Foxl2CreERT2;eYFP* pituitary induced *in utero* (**D**), with the induction timing indicated above. The experiment was performed on at least three independent samples, in both sexes, with a similar result obtained. The scale bars in **A** represents 50 µm and the scale bar in **C** represents 200 µm for both **C** and **D**.

may induce this Leydig cell expansion because, while foetal Leydig cells develop independently of LH, postnatal cells require it.

Altogether these results suggest that GnRH and gonadal steroids, signals extrinsic to the pituitary that are well known for their action on gonadotrophs, are unlikely to play a role in the differentiation of SCs into gonadotrophs. Perhaps it is the abrupt cessation of exposure to the maternal environment that may hold clues to the initiation of this postnatal wave of gonadotroph differentiation and potentially mechanisms of minipuberty initiation. Finally, in addition to helping our understanding of congenital diseases affecting puberty, our findings may explain why the neonatal period is particularly sensitive to exposure to endocrine disruptors[47]. While their effects are often explained by interference with oestrogen signalling, they can also have other, less well-characterized, mechanisms of actions and could thus interfere directly with gonadotroph postnatal emergence.

## Methods
### Mice
All animal experiments carried out were approved under the UK Animals (Scientific Procedures) Act 1986 and under the project licenses n. 80/2405 and PP8826065 and by the Francis Crick Animal Welfare and Ethical Review Body (AWERB). *Sox2[tm1(RFP/rtTA)>rlb]* (this study), *Ednrb[em1(Ednrb-T2A-rtTA)Crick]* (this study), *GT(ROSA)26Sor[tm1(EYFP)Cos]* referred to as *Rosa26[ReYFP]* for Reporter eYFP[48], *(no gene)[Tg(tetO-Cre)LC1Bjd] [49]*, *Lef1[tm1(EGFP/Cre/ERT2)Wtsi]* (EMMA/ Infrafrontier, stock ID EM 10707) and *Foxl2[tm1(GFP/cre/ERT2)Pzg]* (JAX stock #015854) were maintained on C56BL/6Jax background. Mice are housed in isolators with a light cycle of 12 hours day and 12 hour night with a dawn to dusk setting, at a temperature of 22 °C ± 2 °C with a humidity of 55% RH ± 10%.

**Generation of Sox2-T2A-TgRFP-T2A-rtTA mice.** A Sox2-T2A-TgRFP-T2A-rtTA cassette was inserted in the *Sox2* locus by homologous recombination (GenOway) to generate *Sox2[tm1(RFP/rtTA)>rlb]* (Sox2[rtTA]). The neomycin resistance selection cassette was removed. *Sox2[rtTA]* heterozygotes were exclusively used in this study. *Sox2[rtTA/+]* were bred with *(no gene)[Tg(tetO-Cre)LC1Bjd]* and *GT(ROSA)26Sor[tm1(EYFP)Cos]* to perform lineage tracing experiments.

**Generation of Ednrb-T2A-rtTA mice.** The *Ednrb*-T2A-rtA strain was generated by the Genetic Modification Service at the Francis Crick Institute using CRISPR-Cas9 assisted targeting to insert a dox-inducible reverse transactivator into exon 8 of the *Ednrb* gene via a GSG and T2A peptide linker and followed by a stop codon. The donor template contained a 774 bp insert of GSG, T2A and rtTA3, with 692 bp of homology at the 5' end and 700 bp homology at the 3' end. The AAV donor vector was synthesised and packaged into AAV serotype 1 by VectorBuilder. The guide sequence used was 5'-ATAAATA-CAGCTCGTCTTGA-3', that was synthesised as a synthetic guide RNA by IDT[TM]. AAV transduction and delivery of CRISPR Cas9 reagents were performed into 1-cell C57BL/6 J zygotes as previously described[50].

**Cre Induction.** For *Sox2[2A-rtTA/+];TetO-Cre;Rosa26[ReYFP]* and *Ednrb[2A-rtTA/+];TetO-Cre; Rosa26[ReYFP]* induction, two consecutive doses of 9-tert-Butyl

Doxycycline HCl (9TBDox)[31] were administered by subcutaneous injection on two consecutive days between P0 and P4 (125 µg/g body weight). For *Lef1[CreERT2/+];Rosa26[ReYFP/+]* pup induction, a single subcutaneous injection of tamoxifen (0.25 mg/g body weight) was administered. For *Foxl2[CreERT2/+];Rosa26[ReYFP/+]* induction, tamoxifen (0.2 mg/g body weight) was administered to pregnant dam at 11.5 days post coitum and pups fostered at 19.5dpc.

**Cetrorelix treatment.** Pups were administered with cetrorelix acetate (Sigma, C5249) diluted in water every 3 days, or water control, by subcutaneous injection from P0 to P33 (3 µg/g body weight) as described[51].

**Neonatal gonadectomies.** Gonadectomies were performed at P3 in *Sox2[2A-rtTA/+];TetO-Cre; Rosa26[ReYFP]* neonates. Once pups were anaesthetised using isoflurane, a single incision was performed on the skin of the back in females, and in the lower abdomen in males, followed by incisions of the abdominal muscles to reach the gonads. Gonads were then removed in experimental animals, or simply exposed in sham operated controls. Surgical outcomes were optimised by fostering pups with dams from strains displaying greater maternal care, such as outbred CD1.

**Flutamide treatment.** Flutamide (Sigma-Aldrich, F9397) was dissolved in a solution of 10% ethanol in peanut oil to a dosage of 10 mg/ml/kg body weight. To block the effect of the neonatal surge of testosterone, pregnant dams were treated at 19.5 dpc. Treatment was initiated in pups at P0 by subcutaneous injection and continued every three days until P21. Lineage tracing was induced with Dox at P1 and P2. The dosage was chosen based on reports of effective androgen receptor antagonism without significant secondary effects (such as changes in sex organ weight, gonadal development, anogenital distance, early-life body weight) or notable toxicity[36].

### Single-cell sequencing and immunofluorescence
Three *Sox9[iresGFP/+]* male and three *Sox9[iresGFP/+]* female P3 pituitaries were dissociated and cells sorted as described[22]. Quality checks for single-cell suspension and single-cell sequencing were performed as described[22]. Chromium Single Cell 30 Reagent Kits User Guide (v2 Chemistry) was used.

For immunofluorescence on plated cells, freshly dissociated cells were plated on Superfrost plus slides for 90 min in a cell incubator, fixed for 20 minutes on ice in 4% paraformaldehyde (PFA), and immediately processed for immunofluorescence.

### Bioinformatic analyses
Alignment, Seurat analysis, integration and cluster annotation were performed using CellRanger and Seurat. Raw reads were initially processed by the Cell Ranger v.3.0.2 pipeline[52], which deconvolved reads to their cell of origin using the UMI tags, aligned these to the mm10 transcriptome (to which we added the eGFP sequence https://www.addgene.org/browse/sequence/305137/ to detect eGFP expressing cells) using STAR (v.2.5.1b)[53] and reported cell-specific gene expression count estimates. All subsequent analyses were performed in R v.3.6.0[54]

using the Seurat (v3) package[55]. Genes were considered to be 'expressed' if the estimated (log10) count was at least 0.1. Primary filtering was then performed by removing from consideration cells expressing fewer than 50 genes and cells for which mitochondrial genes made up greater than 3 standard deviations from the mean of mitochondrial expressed genes. PCA decomposition was performed, and, after consideration of the eigenvalue 'elbow-plots', the first 20

components were used to construct the UMAP plots per sample. Cluster-specific gene markers were identified using a Wilcoxon rank sum test and the top 20 genes ranked by logFC per cluster were used to generate a heatmap. Clusters were annotated using label transfer methods within Seurat using the GSE120410[27] and GSE217648[22] datasets. Clusters were further annotated using cell-specific signatures (Supplementary Table 6).

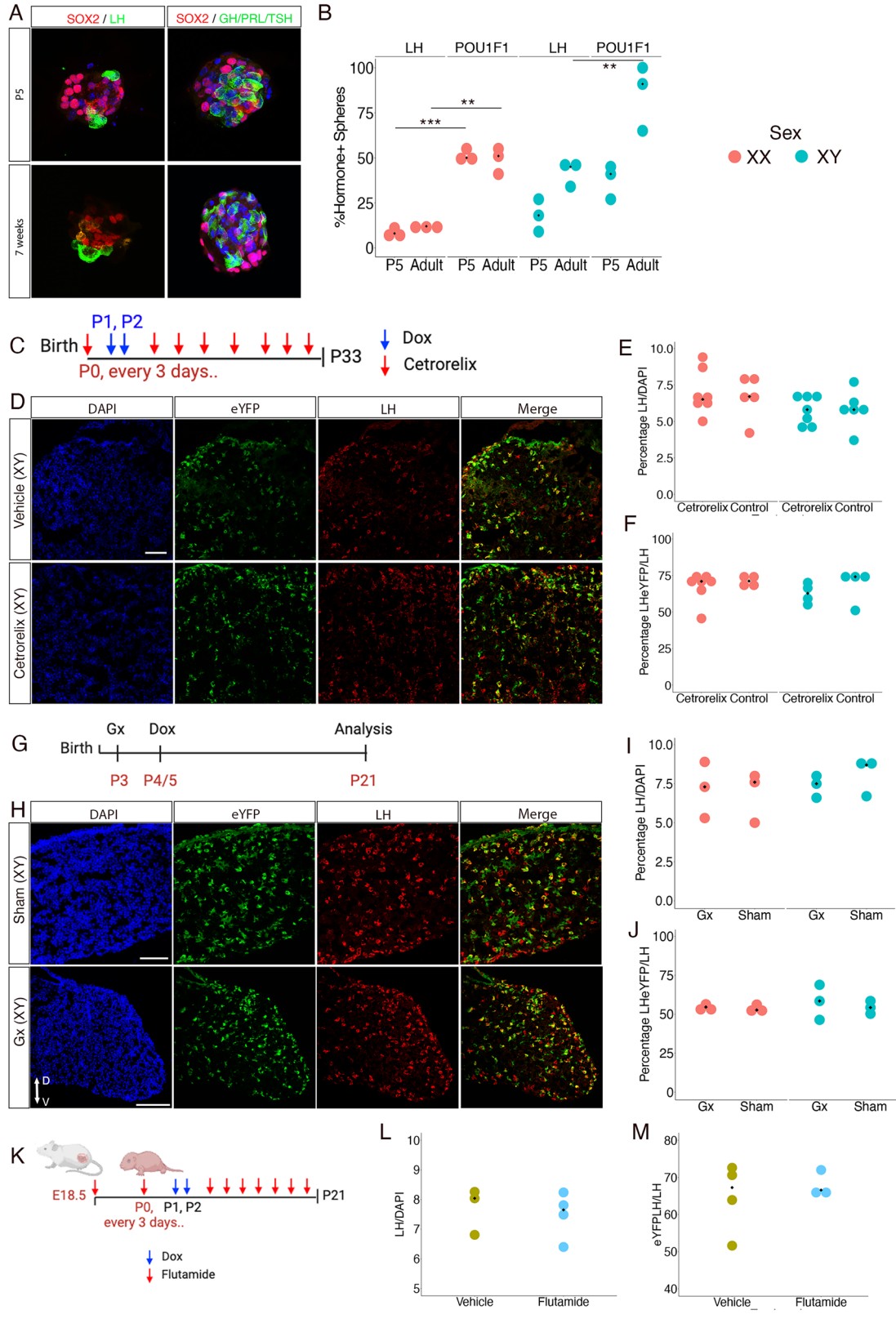

**Fig. 5 | Regulation of neonatal SCs differentiation. A** Differentiation was induced in pituispheres from P5 (top) or 7-week-(bottom) old male pituitaries. The presence of differentiated cells within spheres was evaluated by immunofluorescence using LH or collectively GH, PRL and TSH antibodies for POU1F1 lineage cell detection. SOX2 staining confirms the presence of pituisphere forming-cells. **B** The proportion of spheres containing LH or POU1F1 lineages cells were compared between ages in both sexes. Multiple two-tailed unpaired-tests with Holm-Šídák post hoc test, cultures from $N = 3$ animals per sex and per age (for females, adj.$p = 0.0002$ at P5 and 0.0097 in adults; for males adj.$p = 0.07$ at P5 and 0.004 in adults). **C** Timeline of GnRH antagonist (cetrorelix) administration and lineage tracing induction (Dox) in *Sox2rtTa;eYFP* pups. **D** Immunofluorescence for eYFP and LH in pituitaries of P33 cetrorelix treated male and control mice showing no apparent differences in SC-derived gonadotroph emergence. **E,F** Quantification in both sexes at P33 shows no effect of cetrorelix on either the proportion of LH[+ve]/DAPI[+ve] cells (**E**) or the proportion of eYFP; LH double[+ve]/LH[+ve] cells (**F**). **G** Timeline of gonadectomy and lineage tracing induction (Dox) in *Sox2rtTa;eYFP* pups. **H** Immunofluorescence for eYFP and LH in pituitaries of P21 gonadectomised male and sham-operated control showing no apparent differences in SC-derived gonadotroph emergence. **I–J** Quantification in both sexes at P21 shows no effect of gonadectomies on either the proportion of LH[+ve]/DAPI[+ve] cells (**I**) or eYFP; LH double[+ve]/LH[+ve] cells (**J**). **K** Timeline of androgen antagonist (flutamide) treatment and lineage tracing induction (Dox) in *Sox2rtTa;eYFP* pups. Created in BioRender. Rizzoti, K. (2025) https://BioRender.com/ uge0o92. **L,M** Quantification in males at P21 shows no effect of flutamide on either the proportion of LH[+ve]/DAPI[+ve] cells (**L**) or the proportion of eYFP; LH double[+ve]/LH[+ve] cells (**M**). Scale bar = 80 μm for all images. On graphs, each dot represents one animal with median values indicated. Source data are provided as a Source Data file.

---

Cell trajectories were identified using the package 'Slingshot' (version 1.4.0)[56], using the undifferentiated cluster as a starting point and the PCA co-ordinates. Lineages were identified showing specific trajectories ending in specific differentiated cells. Heatmaps were made using the smoothed expression of differential genes, and the cells are ranked by increasing pseudotime.

In order to investigate transcription factor activity along pseudotime, the package SCENIC[24] was used. A list of 948 putative mouse regulatory binding sites found in promoter regions of expressed genes were used to identify shared regulatory networks. Regulon activity score per transcription factor was calculated using the AUCell function, where the enrichment of target genes was measured using the area under the curve of the target gene relative to the expression-based ranking of all genes. The top 200 regulons ranked by activity score were using to generate a binarized heatmap, where the cells were ranked by pseudotime.

## Pituisphere culture

Dissociated anterior pituitary lobes (see above) were seeded at $50 \times 10^3$ cells/ml in pituisphere medium containing EGF, FGF and 10% serum[34]. After 7 days in culture, pituispheres were transferred into differentiating conditions (on Matrigel-coated coverslips, without growth factor or serum)[34]. Differentiated pituispheres were fixed for 30 minutes in 4% PFA at 4 °C, and immunostaining was performed on the coverslips. Three independent repeats were performed.

## Immunofluorescence, analyses of cell proliferation RIA

Pituitaries were fixed overnight in 4% PFA at 4 °C. Immunofluorescent stainings were performed on cryosections as previously described[57]. The following primary antibodies were used: Goat anti-SOX2 (Biotechne, AF2018), rabbit anti-FOXL2 (gift from Dr. D. Wilhelm), rabbit anti-PIT1 (gift from Dr. S. Rhodes), rabbit anti-CYP17A1 (Abcam, EPR6293), rat anti-GFP (Nacalai Tesque, 04404-84), rabbit anti-LEF1(Abcam, ab137872), and hormone antibodies anti-LH, GH, POMC, and PRL from the National Hormone & Peptide Program (NHPP, previously provided by A.F. Parlow, Torrance, California, USA). Antibodies were typically used at a dilution of 1/500, except anti-LEF1 and anti-CYP17A1 antibodies, both used at 1/250. Staining with anti-LEF1 and anti-PIT1 antibodies required antigen retrieval. Slides were then incubated with Alexa Fluor secondary antibodies. Imaging of stained tissue sections was performed on a Leica SPE microscope and imaging of plated cells was done on an Olympus spinning disk microscope. For all immunofluorescence images shown, representative images are shown from a minimum of $N = 3$ animals for both sexes.

For proliferation analyses, pituitaries were harvested following a one-hour EdU pulse (30 μg/g body weight). EdU incorporation was detected using the Click-iT EdU imaging kit following the manufacturer instructions (Thermo Fisher Scientific). Sections were washed with PBST and mounted using Aqua-Poly/Mount (Polysciences, Inc., Warrington, PA, USA).

## Radio-immuno assay (RIA)

For RIA, the anterior pituitary glands were homogenized in phosphate-buffered saline. Protein contents were then measured and aliquots assayed for GH and LH contents using specific and well-characterised National Hormone and Pituitary Program (NHPP) reagents kindly provided by A.L. Parlow[58].

## Whole mount immunofluorescence

The iDISCO+ protocol was followed, as described (https://idisco.info/wp-content/uploads/2015/04/whole-mount-staining-bench-protocol-methanol-dec-2016.pdf) using Chick anti-GFP (Aves, GFP-1020) and Rabbit anti-FSH from the NHPP. Posterior and intermediate lobes were removed to facilitate antibody penetration. In addition, extension of the primary antibody incubation time up to 2 weeks greatly improved staining detection across the organ. Clearing was performed in a benzyl alcohol/benzyl benzoate 1:2 solution. Samples were then transferred and kept in ethyl cinnamate. Imaging was performed on a Light Sheet Lavision UM II microscope and images reconstructed using the Imaris software.

## Cell Counts

For total anterior lobe cell counts, glands were dissociated as described[22] and live cell counted (Countess 3, Invitrogen).

For dissociated cell counts, slides stained by immunofluorescence were scanned. Automated counts were then performed using QuPath-0.3.2 (https://qupath.github.io/). We routinely counted between 100,000 cells at P5 to 300,000 cells in adults.

SOX2[+ve] cells were counted manually on sections (Fig. 1C) using Fiji because the staining on dissociated cells was not satisfactory. For *Sox2*[2A-rtTA/+]*;TetO-Cre; Rosa26*[ReYFP]*;* or *Lef1*[CreERT2/+]*;Rosa26*[ReYFP/+] eYFP;hormone double[+ve] cell counting, pituitaries were serially sectioned on 5 slides. One slide was immunostained for eYFP and the chosen hormonal marker and all double[+ve] cells manually identified. Manually identified double[+ve] cells were individually imaged on a SPE Leica confocal microscope and images subsequently reviewed to assess co-localisation, and double[+ve] cells scored.

All counting results and tests are presented in supplementary tables S4 and S5.

## Statistical analyses

All statistical analyses were conducted using GraphPad Prism (version 10.2.2).

To compare the proportions of endocrine cells, data were assessed for normality using the Shapiro-Wilk test followed by an unpaired t-test (for parametric data) or the Mann-Whitney U-test (non-parametric data). The Holm-Šídák method (fewer comparisons) or the two-stage step-up method of Benjamini, Krieger, and Yekutieli (many comparisons) was applied to correct for multiple comparisons across sexes and time points. In all cases, statistical significance was set at $p < 0.05$ (or $q < 0.05$ after adjustment). Data are presented as mean ±

SD unless otherwise stated. *$P < 0.05$, **$P < 0.01$, ***$P < 0.001$, ****$P < 0.0001$.

## Reporting summary

Further information on research design is available in the Nature Portfolio Reporting Summary linked to this article.

## Data availability

All data needed to evaluate the conclusions in the paper are present in the paper and/or the Supplementary Materials. The single cell expression data for two samples (Sox9iresGFP PND3 XY and Sox9iresGFP PND3 XX) generated in this study have been deposited in NCBI's GEO database under accession code GSE275746. Both the raw data (FASTQ files) and the processed data (CellRanger outputs) can be found within this accession code. $Sox2^{tm1(RFP/rtTA)>rlb}$ mice are available on request. Source data are provided with this paper.

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

## Acknowledgements

We are grateful for the support from present and past members of the Lovell-Badge laboratory. We are indebted to the Francis Crick Institute platforms for their excellent support and expertise. We especially thank J. Mor from the Biological Research facility for taking excellent care of the animals, S. Wood for performing rodent surgeries, D. Bell from the Advanced Light microscopy, and experts from the Advanced Sequencing, Genetic modification service and Histology platforms. Funding: This work was supported by the Medical Research Council, UK (U117512772, U117562207, and U117570590, R.L-B) and the Francis Crick Institute, which receives its core funding from Cancer Research UK (FC001107, R.L-B), the UK Medical Research Council (FC001107, R.L-B), and the Wellcome Trust (FC001107, R.L-B).

## Author contributions

Conceptualization: K.R. and R.L-B. Methodology: D.S and K.R. Bioinfor-matic analyses: P.C and G.G. Investigation: D.S, K.R, Y.S, J.O, E.B, T.F and C.G. Visualization: D.S and K.R. Supervision: K.R and R.L-B. Writing—original draft: K.R and D.S. Writing—review and editing: K.R, D.S, R.L-B, Ph. M and Pa.M. Philippa Melamed: Ph. M. Patrice Mollard: Pa. M.

## Funding

## Competing interests

The authors declare no competing interests.
