## [Transparent Peer Review file · Nature Communications]

Gonadotrophs have a dual origin, with most derived from early postnatal pituitary stem cells.

Corresponding Author: Dr Karine Rizzoti

Version 0:

Reviewer comments:

Reviewer #1

(Remarks to the Author)

This is an outstanding study from an experimental point of view, including the animal models developed, experimental designs, methods and protocols used in the study, and the volume of data, which together qualify this study for publication in top scientific journals. However, the presentation, interpretation, and discussion of the data is not of the same quality, and the text is quite confusing and sometimes misleading.

Rodents are a well-established animal model for pituitary experiments, but there is a tendency in the manuscript to hide this and present it as general findings for mammals, including humans. Mice should be mentioned in the title. The abstract also makes no mention of the experimental animals used. The first sentence in the Introduction cites minipuberty in humans, rather than describing this phenomenon and timing in mice, which is key information needed to understand the data presented.

I do not agree with the authors that they "demonstrated" the existence of a subpopulation of embryonic gonadotrophs in adult pituitary glands. Their data may "suggest" its existence, but definitely not "demonstrate" the presence of this subpopulation. Sox2 vs Sox2+Sox9 expression is not sufficient to establish the heterogeneity of gonadotrophs. To demonstrate the existence of two subpopulations, the authors should show, not just discuss, the heterogeneity of postnatal gonadotrophs analyzing the entire transcriptomes of two subpopulations using Sox2 vs Sox2+Sox9 information to aid cell clustering. The authors acknowledged that such heterogeneity was not detected by scRNAseq analysis, but cited the wrong study. They probably wanted to quote PMID: 31620083 or PMID: 36093576. The right way to address heterogeneity is spatial transcriptomics. Current equipment has sufficient resolution that could solve this problem. However, this is not my suggestion for the current study to be published, but advice for further studies.

The existence of two subpopulations of gonadotrophs was previously studied. Decades ago, it was established that the majority of neonatal rat gonadotrophs (up to P7 days of age) express functional receptors for melatonin. However, during postnatal development the percentage of melatonin-responsive cells and melatonin receptor gene expression progressively decreased and disappeared in postpubertal animals. In these studies, neonatal cells were thought to be embryonic gonadotrophs (which this study supports), as opposed to adult gonadotrophs that appear after the neonatal period and do not express the melatonin receptor. Therefore, embryonic gonadotrophs were considered a transient population (in contrast to this study) and believed to have a lifespan of about 50 days. In that scenario, it is not surprising that adult gonadotrophs are a homogeneous population in scRNAseq studies.

The authors in the Introduction acknowledged the different patterns of LH and FSH release in response to endogenous GnRH. In the Discussion, they concluded: "double origin and/or different localization of gonadotrophs could explain aspects of their function and regulation". This is not a scientific conclusion based on data, but only further speculation about the existence of two subpopulations in adult animals, which, if they exist, should have a function. There are several other possible explanations, including one reviewed in your reference 38. Briefly, LH secretion represents the release of previously stored LH by regulated exocytosis, whereas FSH predominantly reflects release of de novo synthesized hormone by constitutive exocytosis. Regulated exocytosis requires 1-2 seconds, while constitutive exocytosis requires several hours.

In the Discussion, the authors also stated: "This phenomenon may extend to primates, because a sharp postnatal increase in gonadotroph numbers is observed in rhesus monkeys (37), suggesting its relevance to human physiology." The postnatal

increase in gonadotrophs (and other cell types) exist in all mammals, as shown by qRT-PCR analysis in several published manuscripts, and correlates with an increase in pituitary gland mass. However, this does not clarify which process (proliferation or differentiation of proliferative stem cells) accounts for this increase, and how much is the contribution of embryonic gonadotrophs. Moreover, in Fig. 1B, have you normalized the percentage of cells to pituitary mass during postnatal development? If not, it would be useful to show the pituitary mass profile using the time-points examined. Among other things, that will rule out the impression that the number of thyrotrophs progressively decreases postnatally.

The focus of this paper is on gonadotrophs. In the Abstract, Introduction, and Discussion there is no a single word about other hormone-producing cells. However, Figures 1, 3, 5, S1, S2, S3, S4, S5, and S7, and Tables S1, S3, and S4 contains or are exclusively linked to data on other hormone-producing cells. This is more than 50% of the data presented in this manuscript. I do not understand why these data are included in the manuscript if they do not deserve to be mentioned in the Abstract, Introduction, and Discussion. Furthermore, the postnatal anterior pituitary gland contains folliculostellate cells (which express Sox2 and Sox9), but the authors neglect to address the possible role of stem cells in their formation. I believe the authors have two options: to present and discuss only the gonadotroph data in this manuscript, and to published the data on other pituitary cell types separately, or to properly integrate these data into the manuscript.

The biggest problem with some Figures is the font size: Figs. 2, 3, 5, S2, S5, S7, S8. I do not know what the Journal's policy is on font size, but it should not be bellow 7 for regular reading. To review this manuscript, I had to use a 27-inch monitor, which is not popular these days.

Several Figure Legends needs to be expanded upon. It is useful when the reader can understand the figure data by looking at the figure and reading the legend. For example, Fig. S2 forced you to go to the published MS 20. However, it is the time consuming and you cannot find information there about the age and sex of animals used. In Fig. S5, I am still not sure which panels correspond to the cell type. Probably A and B are interchanged.

Reviewer #2

(Remarks to the Author)

Sheridan et al. have uncovered distinct developmental origins for two subpopulations of pituitary gonadotroph cells. The major population of gonadotrophs arises postnatally. Using time courses together with lineage tracing, the authors show that a major postnatal increase in gonadotroph is due to mobilization of Sox2+ stem cells towards the gonadotroph lineage in contrast to other lineages such as somatotrophs that show similar temporal postnatal expansion but without mobilization from stem cells. Single-cell RNA-Seq confirmed this major contribution of stem cells towards the gonadotroph fate and reveals a sexual dimorphism. The postnatal gonadotrophs were found to be localized mostly on the dorsal aspect of developing pituitary in contrast to the foetal gonadotrophs that occupy the ventral medial part of the gland. The two gonadotrophs cell populations thus show unique distributions in addition to developmental origins. The postnatal gonadotrophs could have differentiated in response to hormonal cues coming from either hypothalamic GNRH or from gonadal steroids. This was tested and found not to be the case. It thus appears that the postnatal expansion of gonadotrophs is dependent on pituitary intrinsic mechanisms. Such mechanisms remain to be identified.

The conclusions of this study are interesting in a few respects. First, by showing postnatal mobilization of pituitary stem cells towards the gonadotroph lineage, this provides a first example in normal development of an involvement of pituitary stem cells as otherwise, pituitary stem cells were only shown to be mobilized in response to end-organ ablation such as after gonadectomy or adrenalectomy. Also, this contrasts with other pituitary lineages that expand postnatally but from already existing pools of progenitors or precursors or directly from differentiated cells. The reasons for this unique mechanism of stem cell mobilization towards gonadotrophs remains to be identified but may underly a population of gonadotrophs that have unique properties within the hypothalamo-pituitary gonadal axis.

Specific Comments

P.9-L.20 9TBDOX should be defined.

P.18, Discussion, beginning of the second paragraph. The statement "this strongly suggests that the physiological context..." is a bit strong given that it turned out to be wrong! It may be more interesting here to state that the physiological context may be responsible for the observed postnatal gonadotroph expansion but that it may also depend on cell/cell interactions within the developing pituitary.

P.20-L.3 The conclusion that expansion of other lineages is explained by "proliferation and/or differentiation of committed progenitors" is too limited in scope. Indeed, committed precursors may also be involved as well as proliferation of differentiated cells. This was shown for the corticotroph lineage where postnatal expansion and adult maintenance is largely due to division of differentiated corticotroph cells (Langlais et al., Mol Endo, 2013).

The last conclusion proposed by the authors (last sentence of Discussion) is not immediately obvious because one may ask how endocrine disruptors may affect neonatal development of gonadotrophs since work in the present paper shows that gonadectomy has no effect on the process! If the authors have in mind a putative effect of excess oestrogens, it may be appropriate to test this idea in their model.

Reviewer #3

(Remarks to the Author)

The manuscript by Sheridan et al. describes the origins of two distinct populations of gonadotroph cells in pituitaries of mice. One emerges embryonically and the second, more abundant, population derives from stem/progenitor cells postnatally. The data are novel and of clear significance to the fields of developmental and reproductive biology. The conclusions are largely supported by the data. A novel mouse model was produced to circumvent previous confounds of lineage tracing approaches using tamoxifen.

Below are some 'major' and 'minor' points for the authors to consider.

Major:

1. Though the data are very compelling, they are also descriptive in nature. As a result, it feels like there are some missed opportunities here. The results inspire several questions that are clearly within the expertise of this group to address. First, does the embryonic gonadotroph population play a role in the development of the postnatal population? This could be addressed by genetically ablating (for example, with DTA) the embryonic population. Second, do the two populations play different functional roles? Admittedly, the second question could take considerable time to address and may therefore be beyond the scope of the current investigation. The first question, however, seems like a reasonable one to address here, especially as no mechanism for expansion/development of the postnatal population has been determined.
2. The Cetrorelix experiment (Fig. 5C-F) is not convincing. While it is clear that the treatment had some effect (given the reduced gonadal weight), it is not obvious that all GnRH signaling was blocked. First, Cetrorelix has a half-life of hours, but the drug was administered only every 3 days. Second, though the immunostaining was not quantified, there is no obvious diminution of the LH signal in the Cetrorelix vs vehicle treated mice in Fig. 5D. Therefore, it is likely that there is still some GnRH action in these experiments. Measuring circulating levels of LH would show whether GnRH action is fully blocked (or not). If it is not fully blocked, the authors cannot conclude that the expansion/development of postnatal gonadotropes is GnRH-independent.
3. Statistical analyses are not routinely employed but should be (for example, in Fig. 1, 2B, 3C-D,F, 4B, 5 (multiple panels)). As a result, it is not always clear what is actually different. This is important as the Ns are uniformly low across experiments. Indeed, in some cases, it is not clear if analyses were done on more than one sample (for example, with most immunofluorescence of tissue sections). The authors need to make explicit the Ns (technical and experimental) in all experiments (in the legends). When 'representative' images are shown, it should be indicated how many times the analyses were done.
4. The Methods are not sufficiently detailed in places to enable replication. For example, there is no information provided on the neonatal gonadectomies; the route(s) of administration of Cetrorelix and flutamide to neonates/pups is not indicated; the section on RIA has no details on the assay(s) used.

Minor:

1. While minipuberty is a clearly defined phenomenon in primates, its definition and timing are less clear in rodents. For example, in male mice, there is a perinatal increase in testosterone. In female mice, there is a transient increase in gonadotrophin secretion around the second week of life. This does not occur in males. Do either or both of these constitute minipuberty? If so, it would suggest a different timing (and mechanism) in the two sexes. Throughout the manuscript (including in the title), it might be safer to simply refer to increases in gonadotrophs postnatally without invoking reference to minipuberty, especially as the authors have not actually determined a mechanism driving expansion of gonadotrophs from SC in either sex. The reference to minipuberty implies that increases in gonadotrophs drive the phenomenon, and this has not been established.
2. Abstract, third line: gonadotrophins regulate gamete 'maturation' rather than 'production' (see also line 10 of the Introduction).
3. Abstract, line 9: replace 'keep' with 'maintain'
4. Introduction, third paragraph, line 2: Make clear that the developmental time point refers specifically to mice.
5. Results, first paragraph, line 11: Perhaps describe the ages that were investigated (even though this is in the figure). As described, there is a sense that more time points may have been investigated than were.
6. Results, first paragraph, line 12: Define 'adult'. Is this week 7?
7. Page 7, last line: As no functional data for SHH or WNT signaling are shown, please be careful not to overinterpret gene expression (correlational) data (similar comment toward the end of page 10).
8. Page 9: The authors indicate that they were able to lineage trace approximately 70% of SOX2+ cells. Perhaps in the Discussion, they should comment on why the labeling is less than 100% efficient and what the implications of 'missing' 30% of the cells might mean with respect to data interpretation.
9. Page 12, sentence starting "Examination of traced...": The text refers to P3, but the figure legend indicates P7. Similarly, in the next sentence, the text references P15, but the legend has P21. Please reconcile.
10. In the same paragraph, there is reference to 80% of the dorsal gonadotrophs being YFP positive. However, this was true for only 2 of 3 animals. The third was closer to 50%.
11. Here, and elsewhere (including in the Methods), please check reference to Rosa26ReYFP/+ mice. Should it be eYFP rather than ReYFP? It is not clear what the R refers to given that Rosa26 is spelled out.
12. Page 12, toward the end of the penultimate paragraph (referencing specific text would be easier with page and line

numbers!), there is reference to supplementary Fig. S8. There is a sex difference in the data that is neither mentioned nor explained.

13. Page 15: When first discussing data in Fig. 5, there is no in-text reference to Fig. 5B.

14. Page 15: Please remove reference to the effects of the withdrawal of maternal estrogen. That may apply in primates but is not relevant in mice. Its inclusion here is misleading. This is also a concern at the end of the Discussion. Given apparent differences in the timing of gonadotrope expansion in male and female mice, it seems unlikely that loss of a maternal signal explains the phenomenon under investigation here.

15. Page 15: when first describing the Cetorelix experiment, refer to the experimental design in Fig. 5C.

16. A few lines later, the authors should also make reference to Fig. 5G when introducing the gonadectomy experiment.

17. In Fig. 5J, the proportion of YFP labeled gonadotropes is lower than in other experiments (closer to 50% here). Is this because of the later timing of dox treatment? Some mention of this is merited.

18. With one exception (Fig. 4C), gonadotropes were labeled with LH. There is a population of FSH+/LH- gonadotropes. It would be interesting to know whether these cells emerge embryonically, postnatally, or both.

19. Fig. S9 legend: 'gonadal hypomorphism' should be 'hypogonadism'. 'Lack of gonadotrophins'—this was not demonstrated experimentally.

Version 1:

Reviewer comments:

Reviewer #1

(Remarks to the Author)

The authors addressed all my concerns raised in initial review.

Reviewer #3

(Remarks to the Author)

The authors have adequately addressed the majority of my prior comments. I still think there was a missed opportunity to determine a role (if any) for prenatal gonadotrophs in the development of the postnatal population by ablating the former in GRIC/DTR mice treated with DTA before the emergence to the latter. The results remain descriptive and not mechanistic, but are nevertheless novel. Mechanistic studies would have revealed what controls the emergence of the postnatal population. I am confident this group will determine these mechanisms in future studies.

REVIEWER COMMENTS

Reviewer #1 (Remarks to the Author):

This is an outstanding study from an experimental point of view, including the animal models developed, experimental designs, methods and protocols used in the study, and the volume of data, which together qualify this study for publication in top scientific journals. However, the presentation, interpretation, and discussion of the data is not of the same quality, and the text is quite confusing and sometimes misleading.

We thank the reviewer for their supportive comment. We have substantially modified our manuscript based on all reviewers' comments and hope it is now much improved. Rodents are a well-established animal model for pituitary experiments, but there is a tendency in the manuscript to hide this and present it as general findings for mammals, including humans. Mice should be mentioned in the title. The abstract also makes no mention of the experimental animals used. The first sentence in the Introduction cites minipuberty in humans, rather than describing this phenomenon and timing in mice, which is key information needed to understand the data presented.

We agree with the reviewer and have now indicated in the abstract that our studies relate to the mouse pituitary. We moreover have included recent references for the murine minipuberty timing (Chachlaki et al, 2022 and Delli et al, 2022) in both the introduction and the discussion.

I do not agree with the authors that they "demonstrated" the existence of a subpopulation of embryonic gonadotrophs in adult pituitary glands. Their data may "suggest" its existence, but definitely not "demonstrate" the presence of this subpopulation. Sox2 vs Sox2+Sox9 expression is not sufficient to establish the heterogeneity of gonadotrophs. To demonstrate the existence of two subpopulations, the authors should show, not just discuss, the heterogeneity of postnatal gonadotrophs analyzing the entire transcriptomes of two subpopulations using Sox2 vs Sox2+Sox9 information to aid cell clustering. The authors acknowledged that such heterogeneity was not detected by scRNAseq analysis, but cited the wrong study. They probably wanted to quote PMID: 31620083 or PMID: 36093576. The right way to address heterogeneity is spatial transcriptomics. Current equipment has sufficient resolution that could solve this problem. However, this is not my suggestion for the current study to be published, but advice for further studies.

We agree with the reviewer that we do not demonstrate transcriptional heterogeneity between the embryonic and postnatal populations, or that they perform different functions. Rather their duality is based on differential origin and localization which undoubtedly proves that there are two subpopulations of gonadotrophs in the adult gland. Thanks to our tracing system we can now distinguish these two populations and

compare them in different ways to assert potential differences, as suggested by the reviewer.

We chose to cite the Constantin et al review because the authors examined data obtained in different models, comprising mice, while the references listed by the reviewer describe experiments conducted in rats which are moreover included in the review. It thus appeared more relevant to our context.

The existence of two subpopulations of gonadotrophs was previously studied. Decades ago, it was established that the majority of neonatal rat gonadotrophs (up to P7 days of age) express functional receptors for melatonin. However, during postnatal development the percentage of melatonin-responsive cells and melatonin receptor gene expression progressively decreased and disappeared in postpubertal animals. In these studies, neonatal cells were thought to be embryonic gonadotrophs (which this study supports), as opposed to adult gonadotrophs that appear after the neonatal period and do not express the melatonin receptor. Therefore, embryonic gonadotrophs were considered a transient population (in contrast to this study) and believed to have a lifespan of about 50 days. In that scenario, it is not surprising that adult gonadotrophs are a homogeneous population in scRNAseq studies.

We thank the reviewer for this interesting comment. As noted by the reviewer and in contrast with the earlier data that are referred to in this comment, we indeed show that gonadotrophs born in the embryo persist in the adult gland. We have now examined *Mtnr1a* and *Mtnr1b* expression and, in agreement with the data described above, do not detect expression of the melatonin receptor A nor B in our P3 Sox9iresGFP positive cell dataset. However, we observe expression of *Mtnr1a* in our unpublished P3 Sox9iresGFP negative cell dataset, again in agreement with the comment above. These rare *Mtnr1a* positive cells do not express detectable levels of *Lhb* or *Fshb* and may more likely represent gonadotroph precursors, probably of embryonic origin since they are present in the Sox9iresGFP negative fraction.

The authors in the Introduction acknowledged the different patterns of LH and FSH release in response to endogenous GnRH. In the Discussion, they concluded: "double origin and/or different localization of gonadotrophs could explain aspects of their function and regulation". This is not a scientific conclusion based on data, but only further speculation about the existence of two subpopulations in adult animals, which, if they exist, should have a function. There are several other possible explanations, including one reviewed in your reference 38. Briefly, LH secretion represents the release of previously stored LH by regulated exocytosis, whereas FSH predominantly reflects release of de novo synthesized hormone by constitutive exocytosis. Regulated exocytosis requires 1-2 seconds, while constitutive exocytosis requires several hours. We agree with this comment. Our observation of a dual origin of gonadotrophs raises the possibility of functional heterogeneity based on the origin of the cells. While differential mechanisms of secretion have been shown, as pointed by the reviewer, these do not exclude further complexity, such as heterogeneity in cell secretory

capacity, recently shown in pancreatic β cells (Peng et al, Nature Metab, 2024), or variability in sensitivity of secretory cell according to the context, seen in somatotrophs (Sanchez-Cardenas et al, PNAS 2010). The next logical step is thus to test the physiological relevance of our observation.

We have now modified our previous sentence to reflect current knowledge on differential regulation of gonadotrophin secretion as follow.

“While mechanisms underlining differential regulation of gonadotrophins secretion have been described (5), the intricacies of LH and FSH differential regulation in response to the same ligand, GnRH, are not yet completely understood ...”

In the Discussion, the authors also stated: “This phenomenon may extend to primates, because a sharp postnatal increase in gonadotroph numbers is observed in rhesus monkeys (37), suggesting its relevance to human physiology.” The postnatal increase in gonadotrophs (and other cell types) exist in all mammals, as shown by qRT-PCR analysis in several published manuscripts, and correlates with an increase in pituitary gland mass. However, this does not clarify which process (proliferation or differentiation of proliferative stem cells) accounts for this increase, and how much is the contribution of embryonic gonadotrophs. Moreover, in Fig. 1B, have you normalized the percentage of cells to pituitary mass during postnatal development? If not, it would be useful to show the pituitary mass profile using the time-points examined. Among other things, that will rule out the impression that the number of thyrotrophs progressively decreases postnatally.

The stem cell lineage tracing analyses convincingly show that the increase in gonadotrophs is due to stem cell differentiation. In addition, the EdU incorporation analyses demonstrate that stem cells proliferate while gonadotrophs do not. We were moreover unable to detect proliferation of FOXL2^{+ve} cells which include potential gonadotroph precursors. Therefore, all together this suggests that the stem cell population expands immediately after birth and gives rise to non-proliferative gonadotrophs until puberty.

Concerning the second part of the question, we agree that it is useful to relate endocrine cell numbers to pituitary growth. We have thus counted whole pituitary cell numbers from P5 to one-year in both sexes (Fig.S1A). This showed that cell numbers increase sharply from P21 to 7-week and up to one-year. Furthermore, we observe that the female pituitary comprises more cells than the male in agreement with its known larger size and weight. To estimate the number of each endocrine cell type/pituitary, we related the percentages we previously obtained to the whole cell count (new panel Fig1.B). This suggests that the numbers of all endocrine cell type increase, except thyrotrophs which remain similar throughout life, rectifying the misleading impression given in panel A and B, that thyrotroph numbers may decrease. This has been discussed in the results.

The focus of this paper is on gonadotrophs. In the Abstract, Introduction, and Discussion there is no a single word about other hormone-producing cells. However,

Figures 1, 3, 5, S1, S2, S3, S4, S5, and S7, and Tables S1, S3, and S4 contains or are exclusively linked to data on other hormone-producing cells. This is more than 50% of the data presented in this manuscript. I do not understand why these data are included in the manuscript if they do not deserve to be mentioned in the Abstract, Introduction, and Discussion. Furthermore, the postnatal anterior pituitary gland contains folliculostellate cells (which express Sox2 and Sox9), but the authors neglect to address the possible role of stem cells in their formation. I believe the authors have two options: to present and discuss only the gonadotroph data in this manuscript, and to publish the data on other pituitary cell types separately, or to properly integrate these data into the manuscript.

Our main result concerning gonadotrophs becomes significant mostly because the development of these cells is strikingly different compared to all the other endocrine cell types. Therefore, the inclusion of data concerning all endocrine cell types is both relevant and necessary. Gonadotrophs are the only cells that do not proliferate despite expanding significantly (Fig.1, Fig.S1) suggesting a different mechanism underlies their amplification. This is explained in Fig.3, S5 and S7 because they are the only cell type featuring a significant postnatal stem cell contribution. In Fig.5, to understand whether neonate stem cells are primed to give rise to gonadotrophs, we must compare their capacity to give rise to these cells to that of other endocrine cell types. Pathways leading to stem cell differentiation however appear to share common steps early on and this is demonstrated by comparing markers and different endocrine trajectories (Fig.S2, S3). In FigS4, it appeared important to verify that our novel lineage tracing tool did not affect GH and LH contents. Therefore, we firmly believe that all the data included are necessary and accordingly, these results are referred to in several instances, both the result and discussion sections.

Regarding folliculo-stellate cells, we have previously demonstrated that in the mouse FS and stem cells are, at least transcriptionally, overlapping (Rizzoti et al, 2023). Examining our P3 dataset we find that the murine marker for FS cells (*Aldoc*) is already present in stem cells at this stage, suggesting that FS cells are already present.

The biggest problem with some Figures is the font size: Figs. 2, 3, 5, S2, S5, S7, S8. I do not know what the Journal's policy is on font size, but it should not be below 7 for regular reading. To review this manuscript, I had to use a 27-inch monitor, which is not popular these days.

We have now insured that the same 12 font was used throughout the manuscript.

Several Figure Legends needs to be expanded upon. It is useful when the reader can understand the figure data by looking at the figure and reading the legend. For example, Fig. S2 forced you to go to the published MS 20. However, it is the time consuming and you cannot find information there about the age and sex of animals used. In Fig. S5, I am still not sure which panels correspond to the cell type. Probably A and B are interchanged.

We have included a table (S6) for cell-type signatures to ease Fig.S2 interpretation. Figure S1 and S5 were re-labelled to make them easier to follow. Precisions were added in the legends, we verified that sex and age were included in all figures. The apparent discrepancy that the reviewer may have noted between Fig.S5 and Table S5 is explained by the fact that graphs show individual animal values and the median value, while tables display the mean. This has now been clearly indicated in all legends.

Reviewer #2 (Remarks to the Author):

Sheridan et al. have uncovered distinct developmental origins for two subpopulations of pituitary gonadotroph cells. The major population of gonadotrophs arises postnatally. Using time courses together with lineage tracing, the authors show that a major postnatal increase in gonadotroph is due to mobilization of Sox2+ stem cells towards the gonadotroph lineage in contrast to other lineages such as somatotrophs that show similar temporal postnatal expansion but without mobilization from stem cells. Single-cell RNA-Seq confirmed this major contribution of stem cells towards the gonadotroph fate and reveals a sexual dimorphism. The postnatal gonadotrophs were found to be localized mostly on the dorsal aspect of developing pituitary in contrast to the foetal gonadotrophs that occupy the ventral medial part of the gland. The two gonadotrophs cell populations thus show unique distributions in addition to developmental origins. The postnatal gonadotrophs could have differentiated in response to hormonal cues coming from either hypothalamic GNRH or from gonadal steroids. This was tested and found not to be the case. It thus appears that the postnatal expansion of gonadotrophs is dependent on pituitary intrinsic mechanisms. Such mechanisms remain to be identified.

The conclusions of this study are interesting in a few respects. First, by showing postnatal mobilization of pituitary stem cells towards the gonadotroph lineage, this provides a first example in normal development of an involvement of pituitary stem cells as otherwise, pituitary stem cells were only shown to be mobilized in response to end-organ ablation such as after gonadectomy or adrenalectomy. Also, this contrasts with other pituitary lineages that expand postnatally but from already existing pools of progenitors or precursors or directly from differentiated cells. The reasons for this unique mechanism of stem cell mobilization towards gonadotrophs remains to be identified but may underly a population of gonadotrophs that have unique properties within the hypothalamo-pituitary gonadal axis.

We thank the reviewer for their interest in our study.

Specific Comments

P.9-L.20 9TBDOX should be defined.

Done

P.18, Discussion, beginning of the second paragraph. The statement “this strongly suggests that the physiological context...” is a bit strong given that it turned out to be wrong! It may be more interesting here to state that the physiological context may be responsible for the observed postnatal gonadotroph expansion but that it may also depend on cell/cell interactions within the developing pituitary.

We agree with this comment and have added a sentence reflecting this at the end of that paragraph (and removed strongly from the initial sentence).

“This suggests that other factors specific to the neonate physiological context are implicated, maybe involving interactions with other pituitary endocrine populations, as recently shown between corticotrophs and somatotrophs (44) ”

P.20-L.3 The conclusion that expansion of other lineages is explained by “proliferation and/or differentiation of committed progenitors” is too limited in scope. Indeed, committed precursors may also be involved as well as proliferation of differentiated cells. This was shown for the corticotroph lineage where postnatal expansion and adult maintenance is largely due to division of differentiated corticotroph cells (Langlais et al., Mol Endo, 2013).

We agree with this comment and have amended our discussion.

“this implies that their expansion is explained by amplification and differentiation of committed progenitors, and proliferation of differentiated cells. In agreement, in the POU1F1 lineage, we observe proliferation of both POU1F1^{+ve};hormone^{-ve} progenitors and POU1F1^{+ve};hormone^{+ve} cells. In the adult, endocrine cells maintain their capacity to divide, and this is enough, at least for corticotrophs, to ensure their turn-over in physiological conditions (43), without an apparent requirement for stem cells (44).”

The last conclusion proposed by the authors (last sentence of Discussion) is not immediately obvious because one may ask how endocrine disruptors may affect neonatal development of gonadotrophs since work in the present paper shows that gonadectomy has no effect on the process! If the authors have in mind a putative effect of excess oestrogens, it may be appropriate to test this idea in their model.

The lack of effect following gonadectomy on neonate stem cell differentiation shows that gonadal steroids and peptides are not responsible for induction of this process. Some endocrine disrupting chemicals (EDCs) mimic the effect of oestrogens, and indeed these would not be expected to affect the process we describe here since gonadectomies do not. However, in many cases EDCs mode of action is not entirely known. Bisphenol A (BPA) for example has both estrogenic and non-estrogenic effects. In rodents, female embryonic gonadotrophs are affected following in utero BPA

exposure (Brannick et al, 2012) while post-natal exposure in boys perturbs reproductive hormone levels (Laerkeholm et al, 2023). This latter study implied that the postnatal period encompassing minipuberty represents a window of sensitivity to EDCs, thus suggesting that the phenomenon we describe may be affected by BPA or other compounds. We have clarified this aspect in our last sentence.

“While their effects are often explained by interference with oestrogen signalling, they can also have other, less-well characterized mechanisms of actions and could thus interfere directly with postnatal emergence of gonadotrophs.”

Reviewer #3 (Remarks to the Author):

The manuscript by Sheridan et al. describes the origins of two distinct populations of gonadotroph cells in pituitaries of mice. One emerges embryonically and the second, more abundant, population derives from stem/progenitor cells postnatally. The data are novel and of clear significance to the fields of developmental and reproductive biology. The conclusions are largely supported by the data. A novel mouse model was produced to circumvent previous confounds of lineage tracing approaches using tamoxifen.

We thank the reviewer for pointing the novelty and significance of our study.

Below are some ‘major’ and ‘minor’ points for the authors to consider.

Major:

1. Though the data are very compelling, they are also descriptive in nature. As a result, it feels like there are some missed opportunities here. The results inspire several questions that are clearly within the expertise of this group to address. First, does the embryonic gonadotroph population play a role in the development of the postnatal population? This could be addressed by genetically ablating (for example, with DTA) the embryonic population. Second, do the two populations play different functional roles? Admittedly, the second question could take considerable time to address and may therefore be beyond the scope of the current investigation. The first question, however, seems like a reasonable one to address here, especially as no mechanism for expansion/development of the postnatal population has been determined.

These are important questions we wish to address in future studies. Deletion of embryonic gonadotrophs has been performed in the lab of U. Boehm using Gnrhr-Cre (GRIC) mice using Rosa26floxSTOP DTA mice (Wen et al, 2010). This study has revealed intricacies in embryonic gonadotroph emergence. However, this system would not be

appropriate to study the role of this population postnatally because postnatal gonadotrophs also express *Gnrhr*, and we would end up ablating both populations. We could potentially drive embryonic gonadotroph ablation using *Foxl2CreERT2*, the same way we have labelled these (Fig.4D), but efficiency of recombination is low, and thyrotrophs would also be affected, compromising experimental outcomes. Although we do not have *Gnrhr-Cre* mice at our institution, combining of *Gnrhr-Cre* with *Rosa26floxSTOP* DTR should enable exclusive ablation of embryonic gonadotrophs, by administering DTA before postnatal gonadotrophs emerge. However, this strategy would not be adequate to perform the converse experiment, getting rid of postnatal gonadotrophs exclusively. Our plan is to devise one system enabling us to perform either embryonic or postnatal gonadotroph ablation. This will require the generation and establishment of novel alleles which we believe is beyond the scope of our current manuscript. We furthermore agree that it will be crucial to address the relevance of the dual origin regarding hormonal output, in different physiological contexts and life stages. This will again require the generation of novel genetic models to label each population independently and will be explored in follow-up studies. Finally, we disagree with the comment that our data is “descriptive in nature”. Our compelling demonstration that gonadotrophs have a dual origin was only possible because a novel lineage tracing system. It was designed to be more efficient and less disruptive of the physiological context and purposely developed to precisely address the contribution of stem cells to this early postnatal period, which remained unexplored. In addition, we have investigated potential mechanisms driving gonadotroph differentiation. These analyses excluded a role for gonadal signals and suggest that GnRH may not be involved, in agreement with the bibliography.

2. The Cetorelix experiment (Fig. 5C-F) is not convincing. While it is clear that the treatment had some effect (given the reduced gonadal weight), it is not obvious that all GnRH signaling was blocked. First, Cetorelix has a half-life of hours, but the drug was administered only every 3 days. Second, though the immunostaining was not quantified, there is no obvious diminution of the LH signal in the Cetorelix vs vehicle treated mice in Fig. 5D. Therefore, it is likely that there is still some GnRH action in these experiments. Measuring circulating levels of LH would show whether GnRH action is fully blocked (or not). If it is not fully blocked, the authors cannot conclude that the expansion/development of postnatal gonadotropes is GnRH-independent.

While Cetorelix is a direct antagonist of the GnRH receptor, it is also known to induce a downregulation of its expression. We believe that this longer-term action may, in contrast with a short-term antagonistic effect, explain the efficient blockade of the HPG axis we observe with our protocol (this has been added to the results). Gonadal weights are significantly reduced in both sexes and the clear reduction in CYP17A1 confirms that steroidogenesis is greatly impaired. While we did not measure LH levels, which, at this stage (one-month-old) are very low, our data is entirely consistent with an efficient

blockade of GnRH action. The unaffected proportion of stem cell-derived gonadotrophs thus strongly argue against a crucial role for GnRH. Because, as the reviewer pointed, we cannot exclude that some GnRH is still active we have indicated this point in more detail in the results (“While we cannot exclude that some GnRH remains active, the effects we observe are consistent with an efficient impairment of its action.”) and the discussion (“While GnRH signalling may not be completely abolished in our experiments, the consequences of the antagonist treatment on gonadal development and morphology demonstrate an efficient inhibition of its action. This implies that GnRH is not a crucial factor for differentiation of pituitary stem cells in gonadotrophs, in agreement with previous observations implying presence of gonadotrophs in absence of GnRH signalling.”).

3. Statistical analyses are not routinely employed but should be (for example, in Fig. 1, 2B, 3C-D,F, 4B, 5 (multiple panels)). As a result, it is not always clear what is actually different. This is important as the Ns are uniformly low across experiments. Indeed, in some cases, it is not clear if analyses were done on more than one sample (for example, with most immunofluorescence of tissue sections). The authors need to make explicit the Ns (technical and experimental) in all experiments (in the legends). When ‘representative’ images are shown, it should be indicated how many times the analyses were done.

We appreciate the reviewer’s comments regarding the need for clarity in statistical analyses and experimental replicates, and we have made several revisions to address these concerns. A description of the statistical analyses employed has been added to the Methods section. For all experiments, a minimum of N=3 animals per sex was used, and representative images were obtained from multiple tissue sections and animals; this information has been explicitly included in the Methods section. Figure legends have also been revised to provide more detail on methodology, explicitly stating the number of replicates and the statistical tests performed. For cell and lineage tracing counts, where including statistical comparisons directly in the figures was challenging, we have presented these data in supplementary tables containing the numerical values and corresponding statistical analyses. References to these supplementary tables have been added to the relevant figure legends for clarity and ease of navigation.

4. The Methods are not sufficiently detailed in places to enable replication. For example, there is no information provided on the neonatal gonadectomies; the route(s) of administration of Cetrorelix and flutamide to neonates/pups is not indicated; the section on RIA has no details on the assay(s) used.

Details and references have been added.

Minor:

1. While minipuberty is a clearly defined phenomenon in primates, its definition and timing are less clear in rodents. For example, in male mice, there is a perinatal increase in testosterone. In female mice, there is a transient increase in gonadotrophin secretion around the second week of life. This does not occur in males. Do either or both of these constitute minipuberty? If so, it would suggest a different timing (and mechanism) in the two sexes. Throughout the manuscript (including in the title), it might be safer to simply refer to increases in gonadotrophs postnatally without invoking reference to minipuberty, especially as the authors have not actually determined a mechanism driving expansion of gonadotrophs from SC in either sex. The reference to minipuberty implies that increases in gonadotrophs drive the phenomenon, and this has not been established.

We agree with the reviewer comment and have replaced minipuberty in the title and amended references to it in the text such as “starting early postnatally and up to puberty”.

2. Abstract, third line: gonadotrophins regulate gamete ‘maturation’ rather than ‘production’ (see also line 10 of the Introduction).

Done

3. Abstract, line 9: replace ‘keep’ with ‘maintain’

Done

4. Introduction, third paragraph, line 2: Make clear that the developmental time point refers specifically to mice.

Done

5. Results, first paragraph, line 11: Perhaps describe the ages that were investigated (even though this is in the figure). As described, there is a sense that more time points may have been investigated than were.

Done

6. Results, first paragraph, line 12: Define ‘adult’. Is this week 7?

This has been amended because 7 week-old animals are not adults.

“At 7-weeks, the percentage of endocrine cells obtained for each population (Fig.1A) matches those described previously in adults (19).”

7. Page 7, last line: As no functional data for SHH or WNT signaling are shown, please be careful not to overinterpret gene expression (correlational) data (similar comment toward the end of page 10).

This has been amended:

“...with a potential involvement for WNT and SHH signalling. “ p7

“WNT signalling is active and potentially important during cell fate acquisition...” p10

8. Page 9: The authors indicate that they were able to lineage trace approximately 70% of SOX2+ cells. Perhaps in the Discussion, they should comment on why the labeling is less than 100% efficient and what the implications of ‘missing’ 30% of the cells might mean with respect to data interpretation.

We amended our sentence in the results:

“Remarkably, comprising up to 80% of eYFP^{ve} cells, and likely more since our lineage tracing is not 100% efficient (Fig.S4), we find that gonadotrophs are the population to which SCs contribute by far the most.”

9. Page 12, sentence starting “Examination of traced...”: The text refers to P3, but the figure legend indicates P7. Similarly, in the next sentence, the text references P15, but the legend has P21. Please reconcile.

Done

10. In the same paragraph, there is reference to 80% of the dorsal gonadotrophs being YFP positive. However, this was true for only 2 of 3 animals. The third was closer to 50%. This was amended to “up to 80%”. In some animals the lineage tracing may be less efficient.

11. Here, and elsewhere (including in the Methods), please check reference to Rosa26ReYFP/+ mice. Should it be eYFP rather than ReYFP? It is not clear what the R refers to given that Rosa26 is spelled out.

R refers to Reporter eYFP in contrast to constitutively fluorescent Rosa eYFP animals. This has now been spelt out in the material and methods and corrected in places where the R was missing.

12. Page 12, toward the end of the penultimate paragraph (referencing specific text would be easier with page and line numbers!), there is reference to supplementary Fig. S8. There is a sex difference in the data that is neither mentioned nor explained.

We apologize for not including page and line numbers. The *Ednrb*^{2A-rtTA/+} allele was very inefficient to trace pituitary stem cells. While we saw a significant difference between males and females, the number of cells counted/animal was low (between 8 and 37). Further experiments are thus required to validate this sex difference. This has now been added to fig.S8 legend.

13. Page 15: When first discussing data in Fig. 5, there is no in-text reference to Fig. 5B.

Done

14. Page 15: Please remove reference to the effects of the withdrawal of maternal estrogen. That may apply in primates but is not relevant in mice. Its inclusion here is misleading. This is also a concern at the end of the Discussion. Given apparent differences in the timing of gonadotrope expansion in male and female mice, it seems unlikely that loss of a maternal signal explains the phenomenon under investigation here.

We have modified our reference about the withdrawal of placental estrogen because, as the reviewer pointed, it may only be relevant in primates.

The only sex difference in gonadotroph emergence was observed at p3, in the single-cell RNAseq dataset, with females displaying a potential delay (Fig.2B). We qualify this as a delay rather than a difference, because in our different quantifications starting at P5, we did not observe any difference between the two sexes (Fig.1 and 3); gonadotrophs are present in comparable numbers and the SC-derived population progresses similarly in

both sexes. Therefore, at this stage of our investigations where we have no evidence for sex difference, we feel that changes accompanying birth, linked to the end of exposure to the maternal milieu, may have a role in emergence of SC-derived gonadotrophs.

15. Page 15: when first describing the Cetorelix experiment, refer to the experimental design in Fig. 5C.

Done

16. A few lines later, the authors should also make reference to Fig. 5G when introducing the gonadectomy experiment.

Done

17. In Fig. 5J, the proportion of YFP labeled gonadotrophs is lower than in other experiments (closer to 50% here). Is this because of the later timing of dox treatment? Some mention of this is merited.

It is very likely to be the case. We have noticed a rapid drop in tracing efficiency postnatally which could be explained by silencing of the Tet controlled Cre transgene, as this has been reported to happen to tetracycline-controlled cassettes. This has now been added in the text.

18. With one exception (Fig. 4C), gonadotrophs were labelled with LH. There is a population of FSH+/LH- gonadotrophs. It would be interesting to know whether these cells emerge embryonically, postnatally, or both.

We examined expression of FSH at different stages in postnatal gonadotrophs but did not notice any obvious difference with LH. Both seemed to be co-expressed from the onset of differentiation (as exemplified in Fig.S6). Because we did not examine this in detail, it was not discussed. We are planning to investigate this aspect more carefully.

19. Fig. S9 legend: 'gonadal hypomorphism' should be 'hypogonadism'. 'Lack of gonadotrophins'—this was not demonstrated experimentally.

This figure and its legend have been amended.